# RCStat: A Statistical Framework of Relative Contextualization in Transformers

## Abstract

Estimating the importance of input tokens and their activations in auto-regressive models is a fundamental requirement in many applications, such as key-value (KV) cache compression and attribution. Prior work computes token importance using attention weights, which are obtained by normalizing the raw attention logits (query-key inner products) with a softmax operation. However, the softmax normalization suppresses the rich information within the attention logits. We introduce **RCStat**, a statistical framework that harnesses the *raw attention logits* via *Relative Contextualization (RC)*–a random variable measuring contextual influence from one subset of tokens to another. We derive computationally efficient bounds on the expected RC and demonstrate its utility in two applications: (i) *KV compression*, where RC-based adaptive thresholding evicts substantial portions of the KV cache with minimal quality loss in token generation; and (ii) *Attribution*, where attention heads with high expected RC yield accurate span-level attribution. Across QA, summarization, and attribution benchmarks, RCStat achieves state-of-the-art performance, improving generation quality by 15–40% and attribution accuracy by 2–16%, all without any model retraining.

## 1 Introduction

The transformer's attention (Vaswani et al., 2017) mechanism encodes contextual relationships between tokens into internal state representations. This involves the raw dot-product similarity scores $\langle q, k \rangle$ of the query and key vectors, followed by a softmax normalization. Post-softmax attention weights are widely used for tasks such as attribution (Yue et al., 2023; Phukan et al., 2024) and memory optimization through Key-Value (KV) cache compression (Ge et al., 2024; Liu et al., 2024a; 2023; Li et al., 2024c). However, such transformation introduces structural bias: it sharpens attention toward dominant tokens while flattening mid-range scores, thereby discarding subtle yet potentially meaningful contextual signals. Fig. 1 visualizes pre-softmax attention logits $(\langle q, k \rangle)$ from generated tokens to the prompt. Prompt tokens that are semantically relevant to the generation, i.e., carrying contextual alignment, consistently obtain higher logit values, while unrelated prompt tokens obtain lower logits. These meaningful differences are evident pre-softmax but are obscured post-softmax, where normalization flattens intermediate scores and skews attention (Xiao et al., 2023) toward a few dominant or structurally favored positions (e.g., ``, ``), referred to as *attention sink* phenomenon (Gu et al., 2024; Xiao et al., 2023).

Such information loss becomes consequential in applications requiring fine-grained relevance estimation use-cases, leading to inaccurate token attribution (Li et al., 2024b), sub-optimal KV-eviction (Ren & Zhu, 2024), etc. While post-softmax weights represent localized attention at a specific layer, we posit that raw logits carry a dual role: they encode not only what the current layer attends to but also preserve upstream interactions, offering a richer statistical substrate for analysis.

Despite this potential, the usage of pre-softmax attention remains largely underexplored, primarily due to the lack of statistical tools and frameworks to extract structured insights from unnormalized logits. This work addresses that gap. We propose a probabilistic formalism that models relevance directly in the logit space, at different levels of granularity, enabling actionable and generalizable utilities across multiple downstream tasks.

The informativeness of pre-softmax logits may not be the same across all attention heads. In fact, it is observed in literature that certain heads—often in the middle layers—demonstrate stronger contextual

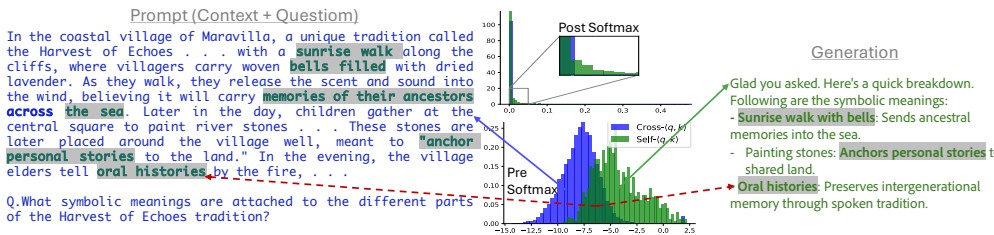

Figure 1: Attribution between generated text (right) and input prompt (left) is analyzed using pre-softmax attention logits vs. post-softmax values. The bottom histogram highlights pre-softmax logits, separating prompt tokens (cross-$\langle$q,k$\rangle$) from generated tokens (self-$\langle$q,k$\rangle$), at $13^{\text{th}}$ layer $23^{\text{th}}$ head from Llama-3B-instruct. Tokens in the overlapping region signify common content, a detail suppressed in the post-softmax histogram above. Using logit distributions, we can attribute (§ 4.2): (1) prompt parts not contextualized during generation (blue arrow), (2) generation parts uninfluenced by the prompt (green arrow), and (3) generation heavily contextualized by the prompt (red dashed arrows).

activity than others (Phukan et al., 2024). Recent interpretability efforts (Dunefsky et al., 2024), such as circuit tracing (Ameisen et al., 2025), offer valuable but often qualitative insights into attention behavior. We take a complementary approach: quantifying how different attention heads behave and using that for KV-cache compression and attribution. For instance, we find most heads have compressible KV-caches, while the few resistant ones provide useful attribution signals.

By operating in the logit space, our goal is to provide a principled and interpretable method for identifying such important heads, facilitating deeper insights into their functional roles. In other words, we propose a method for head-aware attention-logit analysis.

We summarize our contributions as follows:

1. We formalize the notion of contextualization as a set of random variables that capture the relationship between two portions of a text, e.g, a part of prompt and a part of generated tokens, within an attention head. Armed with these random variables, we introduce the relative contextualization (RC) to assign relevance scores at different granularity levels: token-, chunk- and entire text.
2. To estimate the statistics of RC, we derive a easy to compute practical intuitive upper bound and provide an efficient algorithm to compute it, which enables quantitative use of pre-softmax attention logits in downstream tasks.
3. In doing so, we propose RCSTAT, which to the best of our knowledge, is the first unified framework that brings different relevance assignment applications, such as KV-cache compression and token attribution, under one formalism.
4. We demonstrate that RC-based KV compression improves generation quality by 15–40% and achieves 2–5% higher compression over prior SOTA on LLaMa. We also reduce KV cache error by up to 36% on Qwen3 for summarization and QA tasks. For RC-guided attribution, selecting just 2% of attention heads increases span-level attribution accuracy by 2–3% on LLaMa and 13–16% on Qwen3, all without any model retraining.

## 2 RELATED WORK

Interpretability links generation outputs to the input and model internals. Compression and attribution translate this insight into action: the former prunes low-value signals; the latter assigns credit.

**Mechanistic Interpretability of LLMs:** Mechanistic interpretability (Anonymous, 2024) aims to reverse-engineer the internal workings of large language models. Circuit-tracing techniques (Ameisen et al., 2025; Lindsey et al., 2025), such as those from Anthropic, have revealed neuron-level pathways and interpretable MLP circuits. Complementary efforts dissect self-attention heads (Voita et al., 2019), uncovering roles such as induction copying (McDougall et al., 2023; Olsson et al., 2022) and positional tracking (Dufter et al., 2022). Beyond this, representation-level probes in BERT and GPT leverage attribution (Rahimi et al., 2025) and linear classifiers (Du et al., 2025; Rogers et al., 2021; Chanin et al., 2023) to map hidden activations to semantic features. While insightful, these methods often rely on heuristics (Gu et al., 2024) and remain largely qualitative. They lack a unified,

quantitative framework and offer limited direct utility (Zhao et al., 2024). In contrast, our method analyzes the raw attention logits across heads via the statistical lens of Relative Contextualization (RC), to identify which heads are responsible for context-grounding and by how much.

**KV Compression:** KV cache can occupy up to $84\%$ of inference memory for long contexts (Hooper et al., 2024). Prior reduction methods include quantization (Hooper et al., 2024; Liu et al., 2024b; Lin et al., 2024), low-rank approximation (Chang et al., 2024; Dong et al., 2024), state merging (Wang et al., 2024; Agarwal et al., 2025), and eviction (Li et al., 2024a). We focus on eviction, discarding low-value key–value pairs, which can be fixed-size or variable-size. Fixed-size methods enforce uniform budgets per head and layer; early approaches like K-norm (Devoto et al., 2024) and StreamingLLM (Xiao et al., 2023) use vector norms, while attention-based methods, such as SnapKV (Li et al., 2024c), TOVA (Oren et al., 2024), H2O (Zhang et al., 2023), rank tokens by post-softmax weights. QFilter (Godey et al., 2025) isolates signal entries via matrix decompositions, and KVPress (Jegou et al., 2024) applies probabilistic heuristics. Variable-size strategies allow heterogeneous budgets: PyramidKV (Cai et al., 2024) uses a fixed pyramid schedule, and Ada-KV (Feng et al., 2024) adjusts per-layer budgets by token relevance. Manual budgets risk degradation; in contrast, our method sets per-head budgets adaptively via RC.

**Attribution:** Performing attribution is critical for trustworthy generation. Existing methods rely on either gradient signals or post-softmax attention weights. Gradient- and perturbation-driven methods such as Integrated Gradients (Miglani et al., 2023; Sundararajan et al., 2017), LIME/SHAP (Ribeiro et al., 2016; Lundberg & Lee, 2017), and masking/occlusion (Schinagl et al., 2022) trace output sensitivity back to inputs but are computationally expensive. Attention-based methods aggregate post-softmax weights across heads and layers (Abnar & Zuidema, 2020) or formulate attribution metrics (Chefer et al., 2021). However, averaging suppresses medium-strength token associations due to softmax normalization and the presence of sink tokens (Phukan et al., 2024). The learned explainer (Cohen-Wang et al., 2025) mitigates this by training a model to assign reliability scores to heads using labeled data. In contrast, RC computes per-head reliability scores without labels, adapting per example without any extra training.

# 3 Self and Relative Contextualization in LM

To motivate our framework, we represent $\langle q, k \rangle$ as a random variable. This abstraction enables statistical reasoning in scenarios where decisions must be made at a chunk level rather than per token. For example, in KV-compression, eviction decisions must be made prior to generating an entire segment of text, not just individual tokens. Similarly, in attribution tasks, we aim to explain the influence of input tokens over contiguous output spans.

## 3.1 Probabilistic Formalism of Contextualization

Let $V$ be a vocabulary of tokens and $\Omega \subset V^*$ the sample space of all finite token sequences. Let $(\Omega, 2^\Omega, P)$ be a probability space, where the probability measure $P(s)$ of a token sequence $s = (t_1, \ldots, t_n) \in \Omega$ is defined by an auto-regressive pre-trained transformer. For a sequence $s$, let the attention logit be the function $f_s^{l,h} : [n] \times [n] \to \mathbb{R}$, that maps a pair of tokens at positions $i, j$ to

$$f_s^{l,h}(i,j) = \begin{cases} \langle q_j^{l,h}, k_i^{l,h} \rangle & \text{for } i \leq j \leq n \\ -\infty & \text{otherwise,} \end{cases} \tag{1}$$

where $q_j^{l,h}$ and $k_i^{l,h}$ are the query and key vectors of $h^\text{th}$ head in $l^\text{th}$ layer. Our framework develops characteristics for individual heads, but we drop the superscript $^{l,h}$ for brevity henceforth.

A sequence has two parts, $s = p \oplus g$, where $\oplus$ denotes concatenation, $p = \{t_1, \cdots, t_m\}$ is the sequence of prompt tokens and $g = \{t_{m+1}, \cdots, t_n\}$ is the sequence of generated tokens. The demarcation between $p$ and $g$ need not be rigidly associated with their names, prompt, and generation. The $p$ could be a user-given prompt, a conversation history between the user and the language model, or a text that the language model is expected to continue. Similarly, $g$ could be LLM-generated tokens or a portion of an existing text that needs to be analyzed with respect to its previous text $p$. If there is a user question between $p$ and $g$, one may choose to place it at the end of $p$ or the start of $g$.

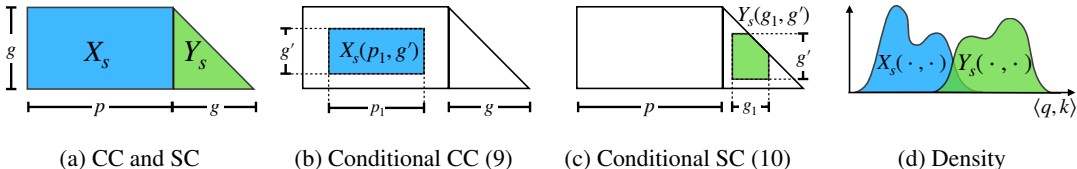

| (a) CC and SC | (b) Conditional CC (9) | (c) Conditional SC (10) | (d) Density |

Figure 2: Illustration of the pre-softmax attention logits $Q^T K \in \mathbb{R}^{|s| \times |s|}$, where $s$ is sequence $s$ with prompt $p$ and generated tokens $g$, as they appear in the last $|g|$ rows the the $Q^T K$ matrix. Fig. 2a shows how the logits in the last $|g|$ rows are partitioned to construct the cross-contextualization (CC) and self-contextualization (SC) random variables. Fig. 2b and 2c show the logits that construct conditional CC and conditional SC. Fig. 2d shows their respective probability density function.

Next, we define four random variables (RVs), all illustrated in Fig. 2, that capture the notion of contextualization at an attention-head level, a sequence level, and a sub-sequence level.

**Definition 3.1** (Cross-Contextualization). We define cross-contextualization of an attention head as

$$F_X(X \le x) := \sum_{s \in \Omega} P(s = p \oplus g) \sum_{t_i \in p} \sum_{t_j \in g} \frac{1}{|p||g|} \mathbf{1}(f_s(i,j) \le x), \tag{CC}$$

where $\mathbf{1}(\cdot)$ is the indicator function. The CC random variable captures the notion of contextualization between the prompt sequence and the generated sequence at a head-level.

**Definition 3.2** (Self-Contextualization). We define self-contextualization of an attention head as

$$F_Y(Y \le y) = \sum_{s \in \Omega} P(s = c \oplus g) \sum_{t_i, t_j \in g} \frac{2}{(|g|+1)|g|} \mathbf{1}(i \le j \ \wedge \ f_s(i,j) \le y). \tag{SC}$$

The (SC) random variable captures the contextualization within the generated tokens at a head-level. Beyond CC and SC, we define conditional counterparts (§A.1) that restrict attention to subsequences of prompt and generated tokens. For $p_1 \subset p$ and $g' \subset g$, conditional CC denoted as $X_s(p_1, g')$ in (9) captures the influence of $p_1$ on $g'$, while for $g_1, g' \subset g$, conditional SC denoted as $Y_s(g_1, g')$ in (10) measures the influence of $g_1$ on subsequence $g'$.

## 3.2 RELATIVE CONTEXTUALIZATION

Due to position embeddings and the auto-regressive nature of generation, self-contextualization values are generally higher than cross-contextualization. Yet some influential prompt tokens also receive higher softmax attention, implying their $\langle q, k \rangle$ values exceed those of certain generated tokens, since softmax is order-preserving. This motivates a third type of variable, *relative contextualization*, designed to isolate prompt-specific influence by subtracting out internal generation bias. Formally, if $X_s(p_1, g) > Y_s(g_1, g)$ with high probability, then the (next-layer) value embeddings of the tokens in $g_1$ become heavily affected by the prompt tokens in $p_1$.

**Definition 3.3** (Relative Contextualization). Assuming a sequence $s = p \oplus g$ is given, and two subsets $p_1 \subset p, g_1 \subset g$, whose complementary tokens $p \backslash p_1$ and $g \backslash g_1$ are given, we define the relative contextualization (RC) random variable as

$$Z_s(p_1, g_1) = \max\left( X_s(p_1, g) - Y_s(g_1, g), 0 \right). \tag{RC}$$

Similar to conditional CC (9) and conditional SC (10) in §A.1, the random variable $Z_s(p, g)$ can be written as $Z|s = \max(X|s - Y|s, 0)$. Estimating the central statistics of RC requires an operation over the joint distribution of $X$ and $Y$. If there are a large number of prompt and generated tokens, then computing any statistics of $Z$ could become prohibitively expensive, as discussed in §3.3. Note, $X_s$ and $Y_s$ are not independent random variables since the key and query vectors of different tokens are intricately dependent on each other.

In the following, we derive a computationally efficient and practical upper bound for RC, without making any distributional assumptions on $X_s$ or $Y_s$. We refer to this upper bound as RC score.

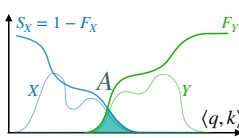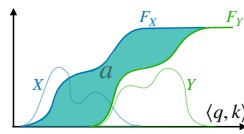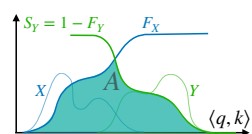

(a) $a \leq \mathbb{E}[\max(X - Y, 0)]$ (b) $\mathbb{E}[\max(X-Y,0)] \leq A$ (c) $a \leq \mathbb{E}[\max(Y - X, 0)]$ (d) $\mathbb{E}[\max(Y-X,0)] \leq A$

Figure 3: Illustration of the upper and lower bounds stated in Theorem 3.4 for two types of relative contextualization: $\max(X - Y, 0)$ and $\max(Y - X, 0)$.

**Theorem 3.4** (Area Under CDFs). *The expected relative contextualization $Z$ is upper bounded by the overlap area $A$ between, a) the area under the marginal CDF $F_Y$ of self-contextualization $Y$, and b) the area under the marginal survival function $S_X$ of cross-contextualization $X$:*

$$\mathbb{E}[Z_s(p_1, g_1)] \leq A_s(p_1, g_1) := \int_{-\infty}^{\infty} \min\left(F_{Y_s(g_1,g)}(t),\ S_{X_s(p_1,g)}(t)\right) dt, \quad (2)$$

*where $S_X(t) = 1 - F_X(t)$, and lower bounded by the area $a$, under $F_Y$ but over $F_X$:*

$$a_s(p_1, g_1) := \int_{-\infty}^{\infty} \max(F_{Y_s(g_1,g)}(t) - F_{X_s(p_1,g)}, 0) dt \leq \mathbb{E}[Z_s(p_1, g_1)]. \quad (3)$$

Our proof of Theorem 3.4 in §A.2 is inspired by (Angelis & Gray, 2021; Vallender, 1974) and uses copula (Durante & Sempi, 2010). Intuitively, the upper bound in (2) captures the area under the overlap region between the values of $X$ and $Y$, as illustrated in Fig Fig. 3b. The upper bound in (2) is tight for continuous CDFs $F_X$ and $F_Y$ (discussed in the proof). However, in our case, they are discrete. An alternative relative contextualization can be defined as $Z'_s(p_1, g_1) := \max(Y_s(g_1, g) - X_s(p_1, g), 0)$ that captures by how much the conditional SC is more than that of the conditional CC. The lower and upper bounds of $Z'_s$ can be formulated similar to $Z$ as in Theorem 3.4, and illustrated in Fig. 3c and 3d.

### 3.3 COMPUTATIONAL COMPLEXITY

We analyze the complexity of directly computing expected RC and computing its upper bound $A_s$ (2) for all the prompt and generated tokens and a given attention head. For direct computation, if we make a simplifying assumption that the joint distribution of $X_s$ and $Y_s$ is a uniform distribution over its discrete support, the expected RC can be calculated as

$$\mathbb{E}[Z_s(p, g)] = \mathbb{E}[Z|s] = \frac{2}{|p||g|^2(|g| + 1)} \sum_{t_i \in p} \sum_{t_j = g} \sum_{t_k \in g} \sum_{t_l \in g:l \geq k} \max(f_s(i, j) - f_s(k, l), 0), \quad (4)$$

using $O(|p||g|^3)$ computations. Similarly, computation of $\mathbb{E}[Z_s(p_1, g_1)]$ requires $O(|p_1||g_1||g|^2)$ computations. An even simpler approximation is to use conditional expectation by assuming output tokens are independent of each other and uniformly distributed: $\mathbb{E}[Z|s] \approx \mathbb{E}_{t_i \sim g}[\mathbb{E}[Z|s, \{t_i\}]] = \mathbb{E}_{t_i \sim g}[\mathbb{E}[\max(X_s(p, \{t_j\}) - Y_s(g, \{t_j\}, 0)]]$.

Although the complexity with this i.i.d. approximation is $O(|p||g|^2)$, which is less than (4), we observe that it performs poorly in downstream tasks such as KV-compression (see §C.6). Hence, we do not use this approximation.

On the other hand, the upper bound $A_s$ can be calculated in $\tilde{O}(|p||g| + |g|^2)$ computations, where $\tilde{O}(n) = O(n \log(n))$, with the Lebesgue integration approach in Algorithm 1. It involves sorting the samples of $X_s$ and $Y_s$, individually in Algorithm 1 and jointly in Algorithm 1 to obtain the unique breakpoints $B$. With appropriate indexing, the complexity of computing CDFs and the minimum values in Lines 7, 8, and 9 becomes linear with respect to the number of midpoints $M$ in Algorithm 1.

---

**Algorithm 1** Area under $\min(F_{Y_s}, 1 - F_{X_s})$

**Require:** Samples of $X_s$ and $Y_s$
**Ensure:** Overlap area $A_s$
1: $X \leftarrow \texttt{sort}(X_s), Y \leftarrow \texttt{sort}(Y_s)$
2: $B \leftarrow \texttt{unique}(\texttt{sort}(X \cup Y))$
3: $L \leftarrow B_{1:\ell-1}, R \leftarrow B_{2:\ell}$     $\triangleright \ell = |B|$
4: $M \leftarrow (L + R)/2$     $\triangleright$ Midpoints
5: $W \leftarrow R - L$     $\triangleright$ Widths
6: **for** each midpoint $m_i \in M$ **do**
7:     $F_X(m_i) \leftarrow \frac{1}{n} \sum_j \mathbf{1}[X_j \leq m_i]$
8:     $F_Y(m_i) \leftarrow \frac{1}{m} \sum_j \mathbf{1}[Y_j \leq m_i]$
9:     $v_i \leftarrow \min(F_Y(m_i), 1 - F_X(m_i))$
10: $A \leftarrow \sum_i v_i \cdot W_i$     $\triangleright \sim$ Lebesgue Integral
11: **return** $A$

---

Similarly, the complexity of computing $A_s(p_1, g_1)$ is $\tilde{O}(|p_1||g| + |g_1||g|)$. Algorithm 1 can be easily extended to compute upper (2) and lower bound (3) for both types of RC in Fig. 3 by modifying Algorithm 1. Although the direct computation of RC in (4) becomes expensive when the number of generated tokens is high; it can be effective when $|g|$ is small. It offers the advantage of computing conditional RC for multiple prompt parts in parallel, which we leverage in the KV-compression task.

# 4 APPLICATIONS OF RELATIVE CONTEXTUALIZATION

Depending on the chosen subsets of the prompt and generated tokens, $p_1 \subset p$ and $g_1 \subset g$, Equation RC can support different applications. In this work, we explore two specific use cases.

## 4.1 KV CACHE COMPRESSION

In an attention head, the value vector $v_j \in \mathbb{R}^d$ for a generated token $t_j$ is computed, using the full KV-cache and under eviction, as:

$$v_j^* = \sum_{i \in [j]} \frac{e^{\langle q_j, k_i \rangle}}{\sum_{i \in [j]} e^{\langle q_j, k_i \rangle}} v_i, \quad \hat{v}_j = \sum_{i \in [j]: i \notin p_e} \frac{e^{\langle q_j, k_i \rangle}}{\sum_{i \in [j]: i \notin p_e} e^{\langle q_j, k_i \rangle}} v_i, \tag{5}$$

where $p_e \subset p$ denotes the set of prompt tokens whose key and value vectors are evicted. An ideal, but combinatorially hard to solve, eviction policy minimizes the degradation in value vector fidelity across all generated tokens $g$ by finding an evictable token set $p_e^*$ such that:

$$p_e^* = \arg \min_{p_e \in 2^p} \frac{1}{|g|} \sum_{t_j \in g} \|v_j^* - \hat{v}_j\|_2. \tag{6}$$

However, $g$ is unknown during decoding. Following SnapKV (Li et al., 2024c), we approximate $g$ by using a window of the last few tokens in the prompt as a proxy $\hat{g}$. By treating expected RC as a score of significance, we decide whether to evict the KV of a token $t_i$, by comparing the scores for the singleton set $p_i = \{t_i\}$ with that of the entire prompt, i.e., evict if

$$\mathbb{E}[Z_p(p_i, \hat{g})] = \mathbb{E}[\max(X_p(p_i, \hat{g}) - Y_p(\hat{g}, \hat{g}))] \leq c \, \mathbb{E}[Z_p(p \setminus \hat{g}, \hat{g})], \tag{7}$$

where $Z_p$ is defined similar to RC, but using the prompt $p$ instead of the entire sequence $s$, and $c$ is a compression hyperparameter. Eviction is adaptive: for a fixed $c$, each head selects a different $p_e$ depending on its contextual load. We observe that a small-sized $\hat{g}$ achieves better performance (see §5.2, hence, use (4) to compute the expected RC, instead of the upper bound (2)). We leave the exploration of other RC formulations in Fig. 3 for score assignment as a future work.

## 4.2 ATTRIBUTION TO CONTEXT TOKENS

Unlike KV-compression, the full token sequence $s = p \oplus g$ is known a priori in the attribution task. In general, the task is to find the spans in $S$ that is most attributable to $g'$, given a generation token span $g' = [t_{j_1}, \cdots, t_{j_2}]$ and a set of spans $S = \{p_1, \cdots, p_{|S|}\}$ from the prompt. A prompt token span $p_i$ could be: a chunk retrieved in the RAG setup, one of the few-shot examples for in-context learning, or a singleton token $p_i = \{t_i\}$ in the input prompt. Our attribution method has three steps.

**Step 1:** Compute the expected RC score $\mathbb{E}[Z_s^h]$ over the full token sequence $s$ for each head $h$ across all layers, then select the top-$k$ heads $\mathcal{H}_k$.

**Step 2:** For each $p_i \in S$ and $h \in \mathcal{H}_k$, compute $\mathbb{E}[Z_s^h(p_i, g')]$, the expected RC from span $p_i$ to generation span $g'$.

**Step 3:** Assign a normalized attribution score $\mathrm{RC}(p_i)$ to each span $p_i \in S$ as:

$$\mathrm{RC}(p_i) = \sum_{h \in \mathcal{H}_k} \frac{\mathbb{E}[Z_s^h(p_i, g')]}{\sum_{p_{i'} \in S} \sum_{h \in \mathcal{H}_k} \mathbb{E}[Z_s^h(p_{i'}, g')]}. \tag{8}$$

Finally, we select the span $p_i$ with the highest $\mathrm{RC}(p_i)$. For long generations, we substitute the expected RC with its efficient upper bound (Eq. 2, visualized in Fig. 3b) computed via Algorithm 1.

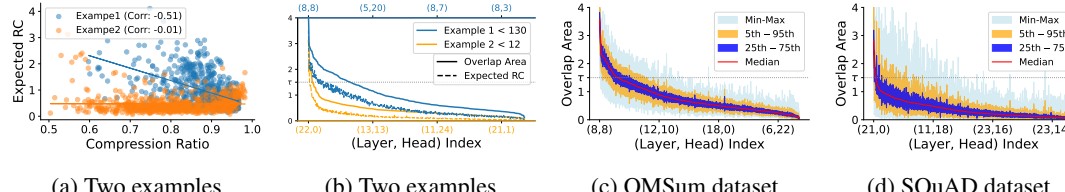

| (a) Two examples | (b) Two examples | (c) QMSum dataset | (d) SQuAD dataset |

Figure 4: All plots are generated using the mentioned datasets with LLaMA-3B model: 28 layers, 24 heads in each layer. Fig. 4b plots the $\mathbb{E}[RC]$ in (4) and its upper bound RC score (2), i.e., the overlap area in Fig. 3b, for all the heads. Fig. 4c and 4d plot the RC scores of all the heads across datasets, and the heads are sorted by their median RC score. Fig. 4a shows how the head-level compression ratio is anti-correlated with the RC score: each point corresponds to a head.

## 5 EXPERIMENTS

We first study the behavior of RC by analyzing its expected scores and upper bounds across attention heads, and then evaluate its utility on two tasks: (i) KV-cache compression and (ii) attribution.

**Datasets and LMs:** For head analysis and KV-cache compression, we use 3 benchmarks: 2000 SQuAD v2.0 (Rajpurkar et al., 2018) (span QA with unanswerable queries), 200 QMSum (Zhong et al., 2021) (query-focused meeting summarization), and 2000 2WikiMultiHop (Ho et al., 2020) (multi-hop RC over linked Wikipedia). For attribution, we use 1300 QuoteSum (Schuster et al., 2023) (summaries with annotated source spans) and 200 VERI-GRAN (Phukan et al., 2024) (grounded-generation attribution). We evaluate 3 LMs from two families: LLaMA-3.2-3B and LLaMA-3.1-8B Instruct (LLaMA; Grattafiori et al., 2024), and Qwen3-8B (Qwen; Yang et al., 2025). (Ref. §8 for details).

### 5.1 ROLES OF ATTENTION HEADS ACROSS TASKS

We analyze whether Relative Contextualization (RC) can distinguish relevant from irrelevant context, differentiate complex summarization from simple QA, and reveal how the number of contextualizing heads varies with the difficulty of the task.

**Two Contrasting Examples:** We provide two examples in §B.1, each with a prompt as the context and the generation as a question-answer pair. In Example 1, the context supports the question; in Example 2, it does not. Fig. 4b shows that head-level RC scores, i.e., the relative contextualization (overlap) between prompt and generation, are uniformly lower for Example 2 with the irrelevant context across all heads and layers. This is reflected in the count of heads whose overlap upper bound exceeds a threshold $\tau = 1.5$; only 12 heads are responsible for contextualization for Example 2, compared to 130 heads for Example 1.

**Sensitivity towards Task Complexity:** Comparing Figures 4c and 4d, we observe that the RC scores are substantially higher for QMSum, complex multi-sentence summarization tasks, than for SQuAD, simple single-hop QA tasks. On average, QMSum requires $9\times$ more heads ($p < 0.05$) than SQuAD, with RC scores higher than $\tau$, indicating that complex tasks recruit a broader ensemble of attention heads. We analyze this phenomenon further in §B.2.

**Consistency of Heads Across Examples:** Figures 4c and 4d also presents the distribution of overlap-area (min–max, 5th–95th, 25th–75th, median) for all 24×28 head-layer pairs on QMSum and SQuAD datasets respective, revealing a pronounced long-tail behavior: a handful of heads exhibit persistently large overlap across inputs, while most heads remain low. In §B.3, we provide detailed head-wise RC distribution plots and heat maps for each dataset and model.

### 5.2 KV CACHE COMPRESSION

**Baselines:** We compare our approach, RCSTAT, against state-of-the-art methods: Knorm (Liu et al., 2024a), SnapKV (Li et al., 2024c), StreamingLLM (Xiao et al., 2023), and TOVA (Liu et al., 2023). Notably, both SnapKV and TOVA operate on post-softmax attention scores.

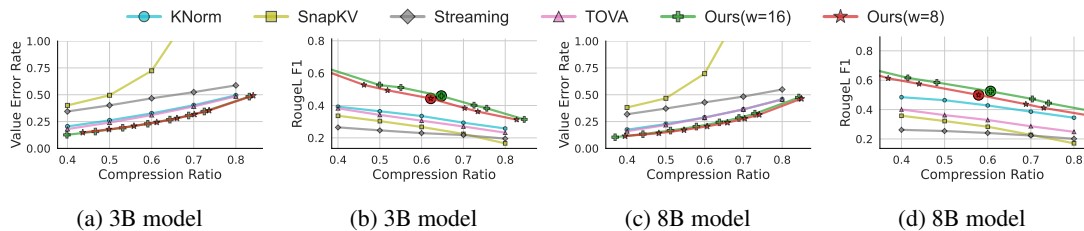

(a) 3B model    (b) 3B model    (c) 8B model    (d) 8B model

Figure 5: KV-cache compression performance on QMSum using LLaMA-3.2-3B and LLaMA-3.1-8B. We report Value Error Rate (VER↓) and generation quality (RL-F1↑) across different strategies.

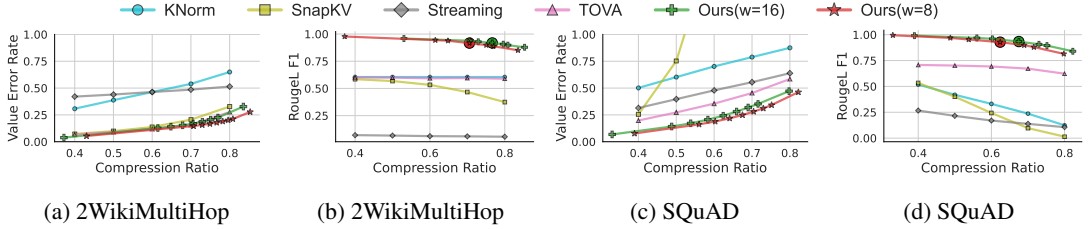

(a) 2WikiMultiHop  (b) 2WikiMultiHop  (c) SQuAD    (d) SQuAD

Figure 6: KV-cache compression results on 2WikiMultiHop and SQuAD v2.0 using LLaMA-3.1-8B.

**Metrics:** The primary metric is the overall compression ratio: the ratio between the evicted KV-cache size and total KV-cache size across all layers and heads. We evaluate the performance using ROUGE-1/L (Lin, 2004) and Value Error Rate (VER), the objective in Eq. (6) normalized by $v_j^*$ inside the summation. For VER, we use the last layer's value vectors of LLM and average across all heads and samples in the dataset. In §C.1, we discuss how VER is more robust to variations in decoding strategies compared to ROUGE, making it a suitable metric for KV-compression. We compare the computational efficiency of our KV compression in §C.4.

**Experimental Setup:** We build upon the baseline implementations provided in the KVPress package (Jegou et al., 2024), using default values for all method-specific hyperparameters. All baselines are evaluated at compression ratios of $0.4, 0.5, \cdots, 0.8$. For RCSTAT, the compression ratio is controlled via the parameter $c$ in (7), where larger values of $c$ result in more aggressive eviction and thus higher compression. We vary $c$ over $0.2, 0.7, 0.8, 1.0, 1.2, 1.3, 1.8$, with the default setting $c{=}1$ highlighted using a circle in all plots. Here, we evaluate our method with window sizes of the last few tokens (see §4.1), with $w \in 8, 16$. §C.5 shows results of further window size variation.

**Generation-Compression Tradeoff:** We present some of our results in Figures 5 and 6. All results for LLaMA models are in §C.2 and for Qwen model in §C.3. We show the results for VER metric in Figures 5a, 5c, 6a and 6c. Across all datasets and models, our method incurs the least VER for all compression ratios. These results show that the fidelity of the internal representation of generated tokens is best preserved in our method. Similarly, results in Figures 5b, 5d, 6b, and 6d, our method achieves the best solution frontier in the trade-off between compression ratio and RougeL F1 score. We notice that, although a lower VER implies higher RougeL, its inverse is not necessarily true: the ordering of solution frontiers of baselines for VER is not the same in RougeL-F1. This is expected, since RougeL measures n-gram output text overlap. In fact, even at $80 \sim 90\%$ compression the an LLM generates answers, not from grounding in the context, but internal model weights learnt during pre-training (Chuang et al., 2024; Feldman et al., 2023; Xu et al., 2024).

**Adaptive Head-wise Eviction:** Our method adaptively evicts KV-caches at the level of individual attention heads. To analyze this, we examine the correlation between head-wise compression ratios and RC scores in Fig. 4a, using the same examples as §5.1. We report only means; variances and other statistics are provided in §C.7. As expected, heads with higher RC scores face less eviction, reflecting their importance for context-grounded generation. In other words, such heads are highly informative and thus less compressible. We leverage these informative heads for attribution.

### 5.3 CHUNK-LEVEL PROMPT ATTRIBUTION

**Baselines:** We focus on training-free, inference-only attribution methods that are computationally efficient for deployment in real production systems. To the best of our knowledge, the zero-train

Table 1: Chunk-level accuracy for attributing extractive spans on QuoteSum and VERI-GRAN ("L3.1" = LLaMa-3.1, "L3.2" = LLaMa-3.2). For LLaMa-3.2-3B and Qwen3-8B, values are pre-softmax. "HS" = best single hidden-state layer baseline.

| Model | QuoteSum | VERI-GRAN |
|---|---|---|
| GPT-3.5 (inline) | 90.18 | 26.40 |
| GPT-4 (inline) | 90.59 | 62.11 |
| BM25 | 75.72 | 68.20 |
| GTR | 72.57 | 53.15 |
| MonoT5 | 89.24 | 67.43 |
| L3.1-8B (HS (Phukan et al., 2024)) | 86.10 | 71.90 |
| L3.1-8B (all heads) | 90.54 | 77.91 |
| L3.1-8B (post-softmax least RC, $k$=20) | 35.72 | 4.69 |
| L3.1-8B (pre-softmax least RC, $k$=20) | 29.49 | 2.81 |
| L3.1-8B (post-softmax most RC, $k$=20) | 90.03 | 71.25 |
| L3.1-8B (**pre-softmax most RC**, $k$=20) | **93.91** | **79.37** |
| L3.2-3B (all heads) | 92.20 | 78.70 |
| L3.2-3B (least RC, $k$=20) | 31.80 | 1.90 |
| L3.2-3B (**pre-softmax most RC,** $k$=20) | **92.70** | **79.00** |
| L3.2-3B (HS; layers 16/14/25) | 86.80 | 71.60 |
| Qwen3-8B (all heads) | 39.10 | 5.60 |
| Qwen3-8B (least RC, $k$=20) | 31.90 | 4.40 |
| Qwen3-8B (**pre-softmax most RC,** $k$=20) | **92.80** | **79.40** |
| Qwen3-8B (HS; layers 16/14/25) | 79.60 | 63.40 |

(a) LLaMa-3.1-8B

(b) Qwen3-8B

Figure 8: Attribution accuracy vs. number of heads. *Most RC*: descending RC; *Least RC*: ascending.

methods include the hidden state (HS) approach (Phukan et al., 2024) and retrieval-based methods, such as BM25 (sparse) (Robertson et al., 2009), GTR (dense) (Ni et al., 2022), and MonoT5 (Nogueira et al., 2020). We also compare against GPT based attribution as done by Phukan et al. (2024).

**Quantitative Results:** We report results for LLaMA3.1-8B in Table 1, On QuoteSum, using the top-$k$ heads improves attribution accuracy by 3% over using all heads, while bottom-$k$ causes a 61% drop. On VERI-GRAN, top-$k$ heads provide a 2% gain, whereas bottom-$k$ heads fall to 2.8% (near random) for LLaMa-3.1-8B. These results confirm the importance of heads with high RC scores computed from pre-softmax attention logits. Interestingly, selecting heads based on post-softmax RC scores performs worse than all heads. We further demonstrate the impact and genaralizability of our technique in identifying top-K RC heads in Table 1 with two more models - a smaller **LLaMA-3.2-3B** and **Qwen3-8B** with different architecture. Attribution using top-k RC heads consistently outperforms baselines, achieving 5.9–13.2% gains over HS on QuoteSum and 7.4–16.0% gains on Veri-Gran. Least-RC heads perform near chance. This effect is particularly pronounced in Qwen3-8B, where attribution drops from 92.8% (most-RC) to 39.1% (all heads), approaching the 31.9% of least-RC heads. LLaMA models show smaller drops (e.g., 93.9→90.5).

**Head-Selection Robustness:** To dive deeper, Fig. 8 shows attribution accuracy as a function of $k$ for LLama-3.1-8B in (a) and Qwen3-8B in (b). When using the top-$k$ heads, accuracy remains near its peak even for small $k$. In contrast, accuracy plummets as more bottom-$k$ heads are included, even up to 800 (out of 1024) heads. This behavior is even more prominent in the Qwen3-8B model as shown in Fig. 7b. This clearly shows that selecting more heads beyond the optimal set, as identified by our technique, can significantly degrade the accuracy. We provide qualitative examples of RC based attribution improvements in §D.

# 6 CONCLUSION

We present RCSTAT, a statistical framework that computes Relative Contextualization (RC) from pre-softmax attention logits with an efficient upper bound, enabling practical KV-cache compression and attribution without retraining; empirically, it achieves state-of-the-art accuracy on both while reducing compute. Analysis shows (1) RC quantifies an LLM's contextualization effort (task difficulty), (2) attention heads exhibit stable, example-agnostic specialization, and (3) the most influential heads cluster in middle layers, consistent with prior work. Explaining *why* and *how* these patterns arise remains future work.

## 7 ETHICS STATEMENT

We have reviewed and comply with the conference Code of Ethics. To the best of our knowledge, this manuscript is original, with all related work properly cited. All authors contributed to the study and take responsibility for its content. There is no undisclosed use of large language models (LLMs); any LLM usage is either integral to the proposed method or explicitly disclosed.

We provide a detailed algorithm to enable reproduction of our results and plan to release source code for full replicability. Comparative results for prior work are taken from their official papers and/or authorized repositories, as specified for each method.

## 8 REPRODUCIBILITY STATEMENT

**Code and Assets.** We use publicly available code, models, and datasets, citepd and listed below with corresponding licenses and versions.

- **Codebase:** We build upon `[KVPress]https://github.com/NVIDIA/kvpress.` *Version:* 3dbf8f4 *License:* Apache 2.0
- **Models:** We use two families of LLMs.
    1. `LLaMA-3` models from (Grattafiori et al., 2024) of two sizes: 3B, *URL:* `https://huggingface.co/meta-llama/Llama-3.2-3B-Instruct`; and 8B, *URL:* `https://huggingface.co/meta-llama/Llama-3.1-8B-Instruct`. *License:* Llama 3.1 and Llama 3.2 Community License
    2. `Qwen` model of 8B size from (Yang et al., 2025). *URL:* `https://huggingface.co/Qwen/Qwen3-8B`. *License:* Apache 2.0.
- **Datasets:** We use datasets as follows:
    - **QMSum** (Zhong et al., 2021) *Version:* Latest GitHub release (accessed 2025-05) *URL:* `https://github.com/Yale-LILY/QMSum` *License:* MIT License
    - **2WikiMultihopQA** (Ho et al., 2020) *Version:* Latest GitHub release (accessed 2025-05) *URL:* `https://github.com/Alab-NII/2wikimultihop` *License:* Apache 2.0
    - **SQuAD v2.0** (Rajpurkar et al., 2018) *Version:* Hugging Face release (accessed 2025-05) *URL:* `https://huggingface.co/datasets/rajpurkar/squad_v2` *License:* CC BY-SA 4.0
    - **QuoteSum** (Schuster et al., 2023) *Version:* GitHub release (accessed 2025-05) *URL:* `https://github.com/google-research-datasets/QuoteSum` *License:* CC BY-SA 4.0
    - **Verifiability-Granular** (Phukan et al., 2024) *Version:* GitHub release (accessed 2025-05) *URL:* `https://github.com/Anirudh-Phukan/verifiability-granular` *License:* CC BY-SA 4.0

    **Environment.** Experiments were conducted using PyTorch 2.1, CUDA 12.1 on A100 80GB, with code and instructions available at `[https://anonymous.4open.science/r/RCStat-289B/README.md]` for reproducibility.

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

## A  FORMALISM OF RELATIVE CONTEXTUALIZATION

### A.1  CONDITIONAL CONTEXTUALIZATION

The CC RV captures the notion of contextualization between the prompt sequence and the generated sequence at a head-level. The same for a given sequence $s$ can be formalized as a conditional RV $X|s$, whose Cumulative Distribution Function (CDF) does not include the outermost summation of CC. The following definition further qualifies CC to a sub-sequence level.

**Definition A.1** (Conditional Cross-Contextualization). Assuming a sequence $s = p \oplus g$ with prompt tokens $p$ and generated tokens $g$ is given, and two subsets $p_1 \subset p$ and $g' \subset g$, whose complementary tokens $p \backslash p_1$ and $g \backslash g'$ are given, we define the conditional CC random variable $X_s(p_1, g')$ as

$$F_{X_s(p_1,g')}(X_s(p_1, g') \leq x) := F_X(X \leq x \mid s, \, p \backslash p_1, \, g \backslash g') = \sum_{t_i \in p_1} \sum_{t_j \in g'} \frac{\mathbf{1}(f_s(i,j) \leq x)}{|p_1| \, |g'|}. \quad (9)$$

The conditional CC (9) represents the influence of prompt tokens in $p_1$ for generating tokens in $g'$. Note, if $p_1 = \{t_i\}$ and $g' = \{t_j\}$ are singleton sets, then the conditional CC $X_s(p_1, g')$ is simply a degenerate random variable at $x = f_s(i,j) = \langle q_j, k_i \rangle$. Similarly, if $p_1 = p$ and $g' = \{t_j\}$, the conditional CC corresponds to the values in $q_j^T K$, where $K \in \mathbb{R}^{d \times |p|}$ is the matrix of key vectors of prompt tokens. Trivially, $X_s(p, g) = X|s$.

We define conditional SC similar to conditional CC in Theorem A.1.

**Definition A.2** (Conditional Self-Contextualization). Assuming a sequence $s = p \oplus g$ is given, and two subsets of the generated tokens $g_1, g' \subset g$, whose complementary tokens $g \backslash g_1$ and $g \backslash g'$ are given, we define the conditional SC random variable $Y_s(g_1, g')$ as

$$F_{Y_s(g_1,g')}(Y_s(g_1, g') \leq y) := F_Y(Y \leq y \mid s, g \backslash g_1, g \backslash g') = \sum_{t_i \in g_1, t_j \in g'} \frac{\mathbf{1}(i \leq j \wedge f_s(i,j) \leq y)}{\sum_{t_i \in g_1, t_j \in g'} \mathbf{1}(i \leq j)}. \quad (10)$$

The conditional SC (10) represents the influences of tokens in $g_1$ for generating the tokens in $g'$ that appear after $g_1$. Trivially, $Y_s(g, g) = Y|s$.

### A.2  THEORETICAL RESULTS

**Theorem 3.4** (Area Under CDFs). *The expected relative contextualization $Z$ is upper bounded by the overlap area $A$ between, a) the area under the marginal CDF $F_Y$ of self-contextualization $Y$, and b) the area under the marginal survival function $S_X$ of cross-contextualization $X$:*

$$\mathbb{E}[Z_s(p_1, g_1)] \leq A_s(p_1, g_1) := \int_{-\infty}^{\infty} \min \left( F_{Y_s(g_1,g)}(t), \, S_{X_s(p_1,g)}(t) \right) dt, \quad (11)$$

*where $S_X(t) = 1 - F_X(t)$, and lower bounded by the area $a$, under $F_Y$ but over $F_X$:*

$$a_s(p_1, g_1) := \int_{-\infty}^{\infty} \max(F_{Y_s(g_1,g)}(t) - F_{X_s(p_1,g)}, 0) dt \leq \mathbb{E}[Z_s(p_1, g_1)]. \quad (12)$$

*Proof.* Following (Vallender, 1974), $\kappa = \mathbb{E}[\max(X - Y, 0)]$ can be written as

$$\kappa = \int_{-\infty}^{\infty} P(X > t \text{ and } Y \leq t) dt \quad (13)$$

$$= \int_{-\infty}^{\infty} \big( P(Y \leq t) - P(X \leq t \text{ and } Y \leq t) \big) dt. \quad (14)$$

Applying Sklar's theorem (Durante & Sempi, 2010), the joint CDF $P(X \leq t \text{ and } Y \leq t)$ can be written as a copula $C$ distribution of marginal CDF values:

$$\kappa = \int_{-\infty}^{\infty} \big( F_Y(t) - C(F_X(t), F_Y(t)) \big) dt. \quad (15)$$

Let $u = F_X(t)$ and $v = F_Y(t)$. Applying Fréchet–Hoeffding bound (Durante & Sempi, 2010),

$$\max(u + v - 1, 0) \leq \quad C(u, v) \quad \leq \min(u, v) \tag{16}$$
$$\Rightarrow \quad v - \min(u, v) \leq v - C(u, v) \leq v - \max(u + v - 1, 0) \tag{17}$$
$$\Rightarrow v - (\min(u - v, 0) + v) \leq v - C(u, v) \leq v - (max(u - 1, -v) + v) \tag{18}$$
$$\Rightarrow \quad \max(v - u, 0) \leq v - C(u, v) \leq \min(1 - u, v). \tag{19}$$

We complete the proof by integrating all sides of (19) w.r.t. $t$. $\qquad\square$

Note, since our CDFs are for discrete random variable, $F_X$ and $F_Y$ are not continuous. Therefore, Sklar's theorem doesn't guarantee a unique coppula $C$ for the $P(X \leq t \text{ and } Y \leq t)$. Moreover, since joint distribution will also be discrete, it will not be invertible. Therefore, we cannot guarantee that the upper bound in Fréchet–Hoeffding bound to be tight.

**Corollary A.3.** *For any small $\delta > 0$, the relative contextualization is upper bounded as*

$$Z_s(p_1, g_1) \leq \frac{A_s(p_1, g_1)}{\delta} \tag{20}$$

*with probability at least $1 - \delta$.*

*Proof.* Let $Z := Z_s(p_1, g_1)$ and $A := A_s(p_1, g_1)$. From Theorem 3.4 we know that $\mathbb{E}[Z] \leq A$.

Now, by Markov's inequality, for any $\delta > 0$, we have:

$$\Pr\left(Z \geq \frac{A}{\delta}\right) \leq \frac{\mathbb{E}[Z]}{A/\delta} \leq \frac{A}{A/\delta} = \delta.$$

Therefore, with probability at least $1 - \delta$, we have:

$$Z < \frac{A}{\delta}.$$

Hence,

$$Z_s(p_1, g_1) \leq \frac{A_s(p_1, g_1)}{\delta}$$

with probability at least $1 - \delta$, as claimed. $\qquad\square$

# B    ANALYSIS OF RC SCORES OF ATTENTION HEADS

## B.1    TWO CONTRASTING EXAMPLES

In the coastal village of Maravilla, a unique tradition called the Harvest of Echoes takes place every October. The event, though relatively unknown outside the region, holds deep cultural meaning for the villagers. The tradition begins with a sunrise walk along the cliffs, where villagers carry woven bells filled with dried lavender. As they walk, they release the scent and sound into the wind, believing it will carry memories of their ancestors across the sea. Later in the day, children gather at the central square to paint river stones with symbols from their family histories. These stones are later placed around the village well, meant to "anchor personal stories to the land." In the evening, the village elders tell oral histories by the fire, including tales of migration, war, lost siblings, and recovered friendships. These stories are not written down but are expected to be retold from memory by the youth in coming years, emphasizing the importance of intergenerational memory. Visitors are welcomed to observe, but only those born in Maravilla may participate fully. The community believes that participation  requires a personal connection to the land, the well, and the stories passed down.

Q.What symbolic meanings are attached to the different parts of the Harvest of Echoes tradition?

G:  Following are the symbolic meanings:

• Sunrise walk with bells: Sends ancestral memories into the sea.

• Painting stones: Anchors personal stories to shared land.

• Oral histories: Preserves intergenerational memory through spoken tradition.

Figure 9: Example 1: Relevant context

In the modern software development landscape, programming languages serve as essential tools that enable developers to create everything from simple scripts to complex systems and applications. Over the years, several programming languages have gained significant popularity due to their versatility, ease of use, performance, and applicability across different domains. Understanding the characteristics and use cases of these languages provides a strong foundation for anyone entering the field of computer science or software engineering.

Below is a detailed overview of some of the most popular and widely-used programming languages in the industry today:

**Python:** Python is a high-level, interpreted programming language renowned for its simplicity and readability, which makes it particularly appealing to beginners. Its clear and concise syntax mimics natural language, allowing developers to write logical code with fewer lines. Python supports multiple programming paradigms, including procedural, object-oriented, and functional programming. One of its key strengths lies in its vast ecosystem of libraries and frameworks, which extend its functionality into areas such as web development (e.g., Django, Flask), data analysis (e.g., Pandas, NumPy), machine learning and artificial intelligence (e.g., TensorFlow, PyTorch), and automation. Its platform independence and active community support have contributed to its rapid rise in popularity, making it one of the most in-demand languages across both academia and industry.

**JavaScript:** JavaScript is a dynamic, high-level scripting language primarily used for client-side web development. Originally developed to enhance interactivity in web browsers, JavaScript has evolved into a powerful and versatile language capable of running on both the client and server sides. Through modern frameworks and libraries such as React, Angular, and Vue.js, JavaScript enables developers to build rich, responsive user interfaces. On the backend, JavaScript is widely used with environments like Node.js, which allow it to be used for server-side programming. The language supports event-driven programming and asynchronous processing, making it ideal for developing web applications that require real-time interaction, such as chat applications and collaborative tools. Its ubiquity across the web and ease of integration with HTML and CSS make JavaScript an indispensable tool in the modern web development stack.

**Java:** Java is a statically-typed, object-oriented programming language that is known for its portability, stability, and scalability. It follows the principle of "write once, run anywhere," meaning compiled Java code can run on any platform that supports the Java Virtual Machine (JVM). Java is a go-to language for building enterprise-level applications, including financial systems, enterprise resource planning (ERP) software, and large-scale web services. Additionally, Java is the primary language used for developing Android applications, making it a cornerstone in the mobile app development ecosystem. Its strong memory management, robust security features, and extensive standard libraries make Java suitable for high-performance applications and distributed systems. Java's long-standing presence in academia and the corporate world continues to ensure its relevance in modern software development.

**C++:** C++ is a powerful, high-performance programming language that extends the C programming language with object-oriented features. It is widely regarded for its fine-grained control over system resources and hardware, which makes it ideal for developing performance-critical applications. C++ is commonly used in system/software development, including operating systems, embedded systems, and real-time systems. It is also prevalent in the game development industry, where low-level memory manipulation and high-speed performance are crucial. Furthermore, C++ is popular in competitive programming and algorithmic challenges due to its speed and flexibility. The language supports both procedural and object-oriented programming paradigms and includes features such as classes, templates, exception handling, and the Standard Template Library (STL), which provides useful data structures and algorithms. While it has a steeper learning curve than some other languages, C++ remains a fundamental language for understanding core computer science concepts and software design.

 In conclusion, these languages—Python, JavaScript, Java, and C++—each offer unique strengths and are suited to different domains and project requirements. Their widespread use and strong community support make them essential tools in the toolkit of any developer.

Q: How to terraform Mars?

G: I think there may be a bit of a problem here! Terraforming Mars is a hypothetical process of making the planet habitable for humans and other Earth-like species. While it's an intriguing concept, it's still largely in the realm of science fiction... for now. Currently, there is no known method or technology that can terraform Mars, and it's unlikely that we'll see significant progress in this area soon. The challenges involved are enormous, including:

1. Distance and communication: Mars is about 140 million miles away from Earth, making it difficult to send resources and personnel.
2. Harsh environment: Mars has a thin atmosphere, extreme temperatures, and toxic chemicals that make it inhospitable to human life.
3. Radiation: Mars lacks a strong magnetic field, which means that radiation from the sun and deep space can be harmful to humans.
4. Gravity: Mars' gravity is only about one-third of Earth's, which can cause a range of health problems for humans.

While there are some concepts and proposals for terraforming Mars, they are still largely theoretical and require significant scientific and technological advancements before they can be considered feasible.

Figure 10: Example 2: Irrelevant context

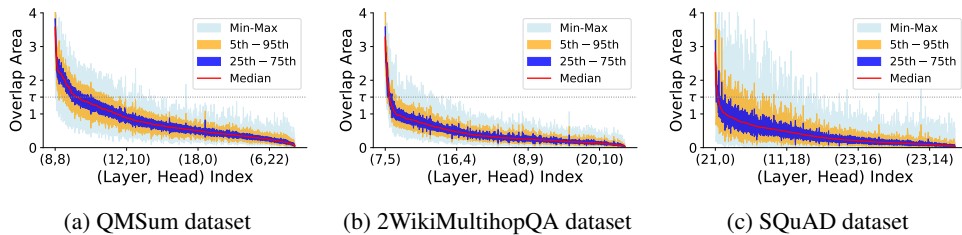

(a) QMSum dataset  (b) 2WikiMultihopQA dataset  (c) SQuAD dataset

Figure 11: All plots are generated using the mentioned datasets with LLaMA-3B model: 28 layers, 24 heads in each layer. All plots show the RC score of the heads (upper bound of expected RC).

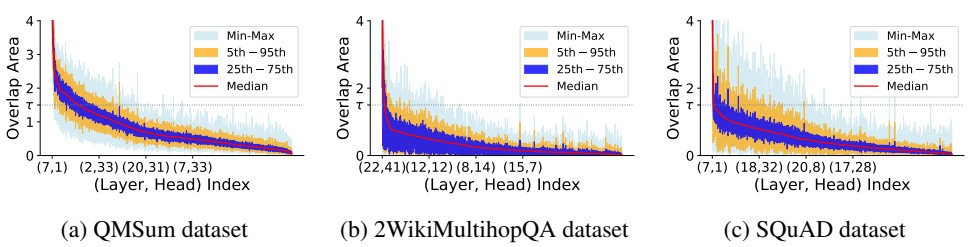

(a) QMSum dataset  (b) 2WikiMultihopQA dataset  (c) SQuAD dataset

Figure 12: All plots are generated using the mentioned datasets with LLaMA-8B model: 32 layers, 32 heads in each layer. All plots show the RC score of the heads (upper bound of expected RC).

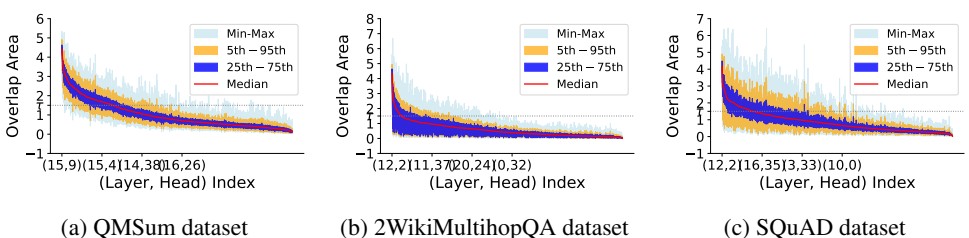

(a) QMSum dataset  (b) 2WikiMultihopQA dataset  (c) SQuAD dataset

Figure 13: All plots are generated using the mentioned datasets with Qwen3-8B model: 36 layers, 32 heads in each layer. All plots show the RC score of the heads (upper bound of expected RC).

## B.2 SENSITIVITY OF RCSTAT TO TASK COMPLEXITY

To better understand the behavior of RCStat across tasks of varying complexity, we compare the RC scores of the heads in LLaMA-3B model in Fig. 11 on single-hop question answering (SQuAD v2), multi-hop question answering (2WikiMultiHopQA), and multi-sentence summarization (QMSum). We apply the same $\tau$-threshold analysis ($\tau = 1.5$) to quantify the number of attention heads with RC $\geq \tau$.

**Single-hop QA (SQuAD v2)**: The median number of heads $\geq \tau$ is 7. For example, in the question "Murders were the base for which story that Capote wrote?", the answer resides in a single span. Only a small number of heads require high RC, reflecting localized retrieval.

**Multi-hop QA (2WikiMultiHopQA):** The median rises to 15, approximately $2\times$ that of single-hop QA. For instance, in "Where was the place of death of the director of the film Happy Hobos?", the model must first identify the director's name and then locate their place of death—two distinct spans that must be aggregated. This leads to a greater number of heads with high RC.

**Summarization (QMSum):** The median further increases to 62, as summarization requires integrating information across the entire document. A large number of heads exhibit high RC, consistent with the need for broad contextualization.

This progression ($7 \rightarrow 15 \rightarrow 62$) highlights that RCStat naturally scales with the breadth of contextualization, ranging from localized lookup (few heads in single-hop), to multi-span aggregation (moderate number of heads in multi-hop), to full-document comprehension (many heads in summarisation).

Importantly, this also demonstrates how RCStat does not require predefining which or how many heads are relevant. Instead, it adapts automatically to the input task, calibrating head selection based on contextual requirements. While RC does not directly measure logical reasoning, it provides a faithful proxy for quantifying the extent of contextual integration needed across tasks of increasing complexity.

### B.2.1 PERSISTENCE OF TOP RC-INFORMATIVE HEADS ACROSS MODELS

Fig. 12 Fig. 13 show similar power law pattern of RC scores for LLaMA-8B and Qwen-8B models across all the datasets, indicating that this a generalizable phenomenon and artifact of model pre-training.

### B.3 RELATIVE CONTEXTUALIZATION DISTRIBUTION: HEAD-WISE ANALYSIS

We show the per-head distribution of relative contextualization, in terms of the upper bound overlap area, in Figures 20, 21, and 22 for QMSum, Squadv2, and 2WikiMultiHop datasets, respectively. The percentile values of these distributions are shown in Figures 23, 24, and 25 respectively. These figures provide empirical evidence of our statement in the conclusion section: "the most influential contextualization heads consistently reside in the model's middle layers, corroborating prior findings." This can also be observed in Figures 4c and 4d in the main paper, where the high-scoring heads correspond to the middle layers: layer indices are shown in the x-axis labels.

## C    EXPERIMENTS FOR KV COMPRESSION

### C.1    EVALUATION UNDER VARIED DECODING CONDITIONS

Robustness to different decoding strategies (e.g., greedy decoding, top-k sampling with temperature, or beam search) is an important consideration for evaluating KV-compression. Conventional textual metrics such as BLEU or ROUGE can be sensitive to these decoding variations.

To address this, our primary metric is VER (Value Error Rate), which directly compares the final-layer hidden states of the response/generated tokens when they are obtained with and without KV eviction. It does not perform auto-regressive decoding; instead, it executes a multi-token forward pass of the ground-truth response tokens, with the KV caches of the prompt tokens evicted. Here, the error cascades across the layers during the forward pass, and the final layer hidden states are compared against the final layer hidden states when there is no KV eviction. This approach decouples compression quality from decoding artifacts, ensuring more reliable comparisons across decoding settings. To capture the error cascading across the auto-regressive decoding, we resort to conventional textual metrics, such as ROUGE-1 and ROUGE-L, using the simple greedy decoding strategy.

**Note:** For the attribution usecase in §5.3, decoding variations are not relevant. Attribution is performed post-generation on the fixed prompt tokens and response tokens as provided by different benchmark datasets.

### C.2    KV COMPRESSION RESULTS ON LLAMA MODELS

This appendix presents all evaluation plots across datasets (QMSum, 2WikiMultiHop, SQuAD) and model sizes (3B, 8B). Each plot compares performance metrics–VER, ROUGE-1 F1, and ROUGE-L F1– across different configurations. A shared legend is included for clarity, and plots are grouped by metric and model size for visual consistency.

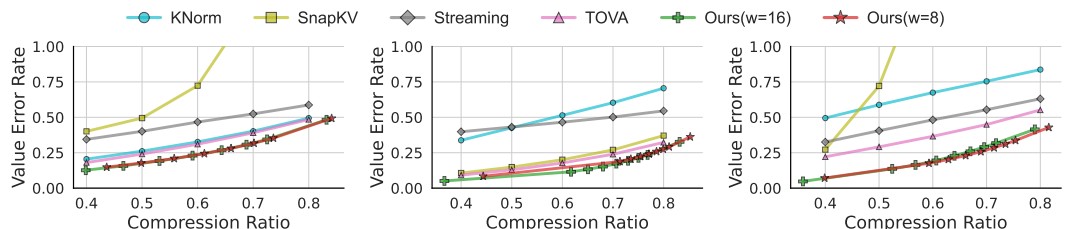

Figure 14: VER scores for 3B model on QMSum, 2WikiMultiHop, and SQuAD datasets.

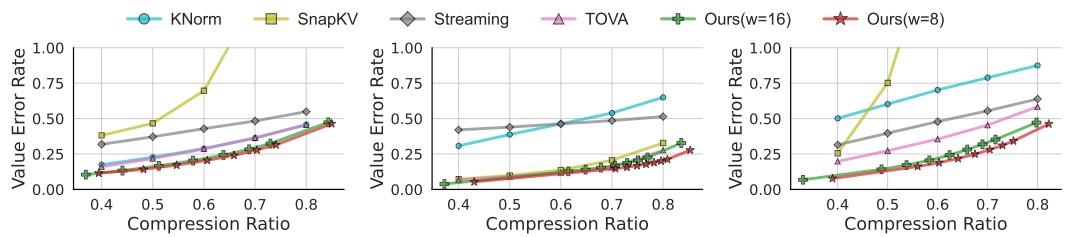

Figure 15: VER scores for 8B model on QMSum, 2WikiMultiHop, and SQuAD datasets.

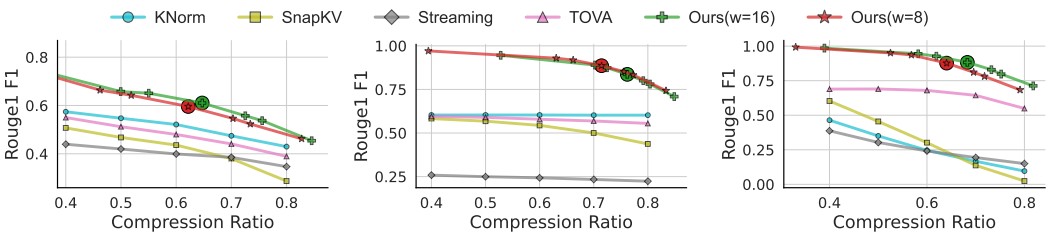

Figure 16: ROUGE-1 F1 scores for 3B model on QMSum, 2WikiMultiHop, and SQuAD datasets.

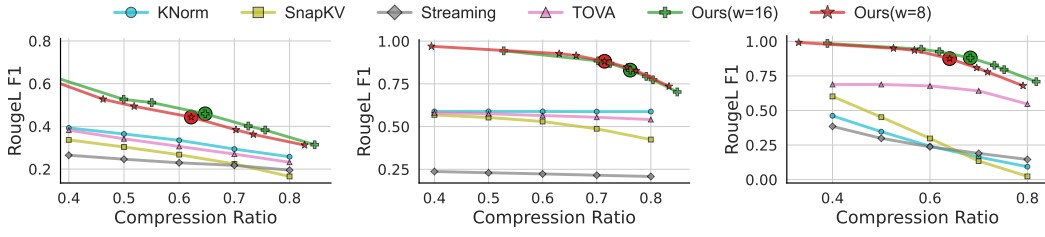

Figure 17: ROUGE-L F1 scores for 3B model on QMSum, 2WikiMultiHop, and SQuAD datasets.

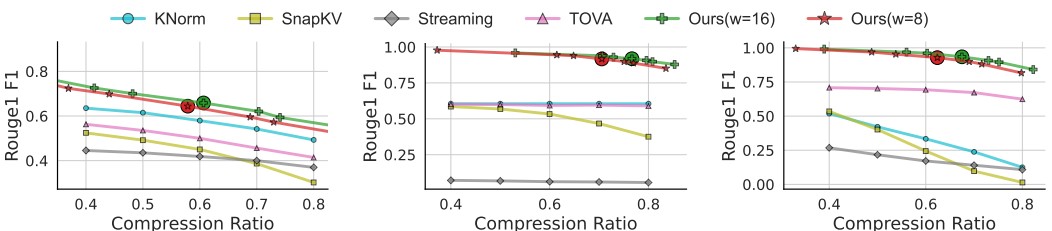

Figure 18: ROUGE-1 F1 scores for 8B model on QMSum, 2WikiMultiHop, and SQuAD datasets.

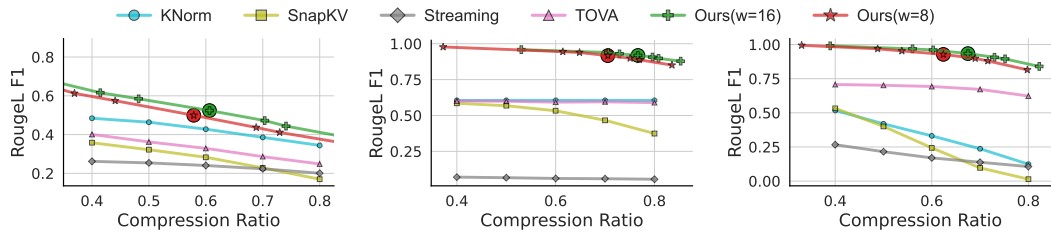

Figure 19: ROUGE-L F1 scores for 8B model on QMSum, 2WikiMultiHop, and SQuAD datasets.

Table 2: **Value Error Rate (VER)** on the QMSum dataset across different compression ratios (50%, 60%, 70%) for LLaMA-3.2-3B and LLaMA-3.1-8B Instruct models. **RCStat (IOT)** assumes *Independent Output Tokens*, while **RCStat (Non-IOT)** does not assume any independence. Here, lower is better.

| Model | Method | 50% | 60% | 70% |
|-------|--------|-----|-----|-----|
| **3B** | TOVA | 0.2408 | 0.3103 | 0.3905 |
| | RCSTAT (IOT) | 0.1956 | 0.2648 | 0.3402 |
| | RCSTAT (Non-IID) | **0.1571** | **0.2295** | **0.3066** |
| **8B** | TOVA | 0.2177 | 0.2859 | 0.3639 |
| | RCSTAT (IOT) | 0.1615 | 0.2290 | 0.3007 |
| | RCSTAT (Non-IID) | **0.1043** | **0.2034** | **0.2836** |

### C.3 GENERALIZABILITY OF RCSTAT ACROSS MODEL FAMILIES

To demonstrate that RCStat is not specific to the LLaMA series, we repeated core evaluations on the **Qwen3-8B** model. Results on the QMSum summarization task confirm that Relative Contextualization properties hold across different model families. RCStat achieves up to **36% lower VER** than TOVA and over **70% lower** than KNorm/SnapKV, validating strong generalization. Similar trends are observed on SQuAD and 2Wiki, where RCStat consistently outperforms all baselines across compression ratios, further establishing its robustness beyond a single dataset.

**Note:** RCStat uses an adaptive compression ratio, but interpolated VER scores at fixed CRs are reported here for comparability with the Figures presented in the main paper.

### C.4 KV-COMPRESSION EFFICIENCY ANALYSIS

In addition to accuracy under fixed compression ratios, efficiency is a critical factor in evaluating KV-compression methods. Efficiency can be viewed along two complementary axes: **memory savings** and **computational latency**.

Table 3: Value Error Rate (VER) with Qwen3-8B for different KV compression methods across three datasets. RCStat consistently shows strong performance across Compression Ratios (CR).

| | | QMSum | | | | | 2Wiki | | | | | SQuAD v2 | | | | |
|---|---|---|---|---|---|---|---|---|---|---|---|---|---|---|---|---|
| Method | CR | 0.40 | 0.50 | 0.60 | 0.70 | 0.80 | 0.40 | 0.50 | 0.60 | 0.70 | 0.80 | 0.40 | 0.50 | 0.60 | 0.70 | 0.80 |
| RCStat ($w = 16$) | | **0.12** | **0.16** | **0.22** | **0.31** | **0.47** | **0.13** | **0.18** | **0.24** | **0.32** | **0.46** | **0.09** | **0.13** | **0.18** | **0.26** | **0.39** |
| RCStat ($w = 8$) | | **0.13** | **0.16** | **0.21** | **0.29** | **0.45** | **0.14** | **0.19** | **0.25** | **0.33** | **0.48** | **0.10** | **0.14** | **0.19** | **0.27** | **0.41** |
| TOVA | | 0.19 | 0.21 | 0.25 | 0.33 | 0.49 | 0.20 | 0.24 | 0.29 | 0.37 | 0.53 | 0.16 | 0.19 | 0.23 | 0.30 | 0.46 |
| KNorm | | 0.39 | 0.41 | 0.48 | 0.61 | 0.80 | 0.40 | 0.45 | 0.52 | 0.65 | 0.85 | 0.32 | 0.37 | 0.44 | 0.58 | 0.77 |
| SnapKV | | 0.45 | 0.55 | 0.72 | 1.34 | 2.33 | 0.47 | 0.58 | 0.75 | 1.30 | 2.25 | 0.41 | 0.52 | 0.69 | 1.20 | 2.10 |
| Streaming | | 0.26 | 0.31 | 0.37 | 0.43 | 0.54 | 0.25 | 0.30 | 0.36 | 0.44 | 0.56 | 0.21 | 0.27 | 0.34 | 0.41 | 0.51 |

**Memory savings.** KV memory footprint is often the dominant bottleneck in real-world deployments. Compression that maintains answer quality while reducing memory is, therefore, a central challenge. RCStat achieves significant improvements in this dimension, providing up to 32% higher memory savings than the best-performing baseline at equivalent accuracy levels (Figure 6c).

**Computational latency.** Latency for compression-based decoding can be decomposed into:

1. **Prefill:** This step remains unchanged across all methods and dominates runtime ( 3 ms per layer on Llama-3.2-3B for QMSum with $\approx 1000$ average prefill length).

2. **Compression decision:** RCStat requires 0.68 ms per layer, which is on par with SNAP (0.66 ms) and TOVA (0.65 ms), while remaining only modestly slower than KNorm (0.13 ms) and STREAM (0.15 ms). Since this step can be pipelined with layer-wise prefill computation, the effective overhead is negligible.

3. **Decode:** Latency improvements during decoding scale directly with compression ratio. Because RCStat maintains higher answer quality even under more aggressive compression, it enables proportionally greater decoding speed-ups.

Taken together, these results show that RCStat offers the strongest balance of memory efficiency and computational speed, outperforming prior methods on memory savings while achieving comparable per-layer latency to state-of-the-art baselines. We will release a head-wise compression implementation to support efficient decoding with RCStat in future work.

## C.5 Effect of Window Size on KV Compression

In KV cache compression, the "window" refers to the last few prompt tokens used as a proxy for future tokens during inference. It is important to clarify that this is not a sliding window. We evaluated window sizes $w \in \{1, 2, 4, 8, 16, 32, 64\}$ on the Llama3.1-8B model across different compression ratios (CR), measuring Value Error Rate (VER) as in Fig. 5c. For each window size, the compression factor in Equation (7) was varied from 0.02 to 2 to obtain different CRs.

We observe from the results in Table 4 that window sizes $w = 8$ and $w = 16$ consistently lie on the Pareto frontier, achieving a balance between compression and accuracy. Smaller windows ($w = 1, 2$) lead to higher VER despite higher compression, while larger windows ($w \geq 32$) yield lower compression ratios. This suggests a "sweet spot" in window size that balances context coverage and relevance.

Table 4: VER under varying window sizes ($w$) and compression ratios (CR) for KV cache compression. Bold numbers indicate Pareto-optimal results.

| | QMSum | | | | | | SQuAD v2 | | | | | | 2WikiMultiHopQA | | | | | |
|---|---|---|---|---|---|---|---|---|---|---|---|---|---|---|---|---|---|---|
| $w$ | 0.4 | 0.5 | 0.6 | 0.7 | 0.8 | 0.9 | 0.4 | 0.5 | 0.6 | 0.7 | 0.8 | 0.9 | 0.4 | 0.5 | 0.6 | 0.7 | 0.8 | 0.9 |
| 1 | - | - | - | - | - | 0.74 | - | - | - | - | - | 1.00 | - | - | - | - | - | 0.92 |
| 2 | - | - | - | - | 0.60 | 0.63 | - | - | - | - | 0.51 | 0.63 | - | - | - | - | - | 0.48 |
| 4 | - | - | 0.27 | 0.36 | 0.42 | 0.52 | - | - | 0.25 | 0.29 | 0.44 | 0.60 | - | - | 0.14 | 0.19 | 0.24 | 0.36 |
| 8 | 0.12 | **0.15** | 0.20 | **0.28** | 0.40 | 0.51 | - | 0.13 | 0.19 | 0.28 | 0.42 | 0.59 | 0.05 | 0.07 | 0.11 | 0.14 | 0.20 | 0.35 |
| 16 | **0.11** | 0.16 | 0.22 | **0.28** | 0.42 | 0.54 | **0.10** | 0.15 | 0.22 | 0.33 | 0.45 | 0.61 | **0.04** | 0.08 | 0.12 | 0.17 | 0.27 | 0.42 |
| 32 | 0.14 | 0.19 | 0.26 | 0.34 | 0.43 | - | 0.21 | 0.29 | 0.38 | 0.47 | - | - | 0.13 | 0.18 | 0.24 | 0.31 | 0.38 | - |
| 64 | 0.13 | 0.19 | 0.26 | 0.35 | - | - | 0.22 | 0.31 | 0.40 | - | - | - | 0.14 | 0.19 | 0.25 | 0.32 | - | - |

## C.6 INDEPENDENCE ASSUMPTION OF GENERATED TOKENS

The result in Table 2 shows that the fidelity of value vectors is higher when RCSTAT is executed without assuming independence for the random variables corresponding to $\langle q, k \rangle$ of generated tokens. Nonetheless, even with the independence assumption, RCSTAT outperforms TOVA, which is the best-performing method in our experiments for the main paper.

## C.7 COMPLETE EXPERIMENTAL STATISTICS FOR KV-COMPRESSION RESULTS

Please find the results of Value Error Rate (VER) inside the VER folder. For the baseline methods, the mean and standard deviations of VER for different compression ratios are saved in csv files with the naming format `<dataset>_<model>_baseline_df.csv`, where the dataset field can be 2WikiMultiHop, QMSum, or SQuAD, and the model field can be 3b or 8b. Similarly, the mean and standard deviations of VER and the mean and standard deviations of the compression ratios for different threshold multipliers are saved in csv files with the naming format `<dataset>_<model>_proposed_df.csv`. Similarly, the results for Rouge1 and RougeL can be found in the All_Rouges folder.

# D QUALITATIVE COMPARISON OF ATTRIBUTION STRATEGIES

To better understand the effectiveness of various attention-based attribution strategies, we compare three different approaches using attention maps from Layer 15 of our model: (1) the mean attention across all heads, (2) the top-scoring head according to our attribution scoring technique, and (3) the worst-scoring head by the same measure. All methods were evaluated on the same input setup: a sales report document with the question "*What were the product sales on November 21st?*" and the answer "*The product sales on November 21st were $177.00.*"

Figure 26 presents the attention heatmaps produced by each of the three strategies. The top-scoring head (Head 30, Fig. 26a) yields a sharply focused attribution map, precisely attending to tokens corresponding to the correct numerical value. In contrast, the mean attention across all heads (Figure 26b) produces a reasonable heatmap but also attends to several unrelated tokens, leading to less interpretable attributions. Finally, the worst-scoring head (Head 16, Fig. 26c) demonstrates diffuse and uninformative attention, highlighting mostly irrelevant tokens.

These observations qualitatively validate our scoring technique for identifying high-quality attribution heads and demonstrate that selectively using the best attention heads can significantly improve interpretability.

# E PRELIMINARIES

Let $X = [x_1, x_2, \ldots, x_T] \in \mathbb{R}^{T \times d}$ be the input sequence of length $T$, where each $x_i \in \mathbb{R}^d$ is an input token embedding and $d$ is the model's hidden dimension. For each attention head $h \in \{1, \ldots, H\}$ in layer $\ell \in \{1, \ldots, L\}$, define

$$Q^{(\ell,h)} = XW_Q^{(\ell,h)}, \quad K^{(\ell,h)} = XW_K^{(\ell,h)}, \quad V^{(\ell,h)} = XW_V^{(\ell,h)}, \tag{21}$$

where $Q^{(\ell,h)}, K^{(\ell,h)}, V^{(\ell,h)} \in \mathbb{R}^{T \times d_h}$ and $d_h = d/H$.

**Attention Logits.** The pre-softmax attention logits are given by

$$Z^{(\ell,h)} = Q^{(\ell,h)}(K^{(\ell,h)})^\top \in \mathbb{R}^{T \times T}, \quad z_{i,j}^{(\ell,h)} = \langle q_i^{(\ell,h)}, k_j^{(\ell,h)} \rangle. \tag{22}$$

**Attention Weights.** The post-softmax weights are

$$A^{(\ell,h)} = \text{softmax}\left(\frac{Z^{(\ell,h)}}{\sqrt{d_h}}\right), \quad a_{i,j}^{(\ell,h)} = \frac{\exp(z_{i,j}^{(\ell,h)}/\sqrt{d_h})}{\sum_{j'} \exp(z_{i,j'}^{(\ell,h)}/\sqrt{d_h})}. \tag{23}$$

**Attention Output.** The output of the head is

$$Y^{(\ell,h)} = A^{(\ell,h)} V^{(\ell,h)} \in \mathbb{R}^{T \times d_h}, \tag{24}$$

and the full multi-head output is

$$\mathrm{MHA}^{(\ell)}(X) = \left[ Y^{(\ell,1)} \| \dots \| Y^{(\ell,H)} \right] W_O^{(\ell)} \in \mathbb{R}^{T \times d}. \tag{25}$$

We refer to $Z^{(\ell,h)}$ and $A^{(\ell,h)}$ as the pre-softmax affinities and post-softmax attention weights, respectively. While $A^{(\ell,h)}$ normalizes attention for token interaction, it also suppresses informative patterns in $Z^{(\ell,h)}$. Our work leverages $Z$ directly to extract statistical signals useful for contextual analysis, KV compression, and attribution.

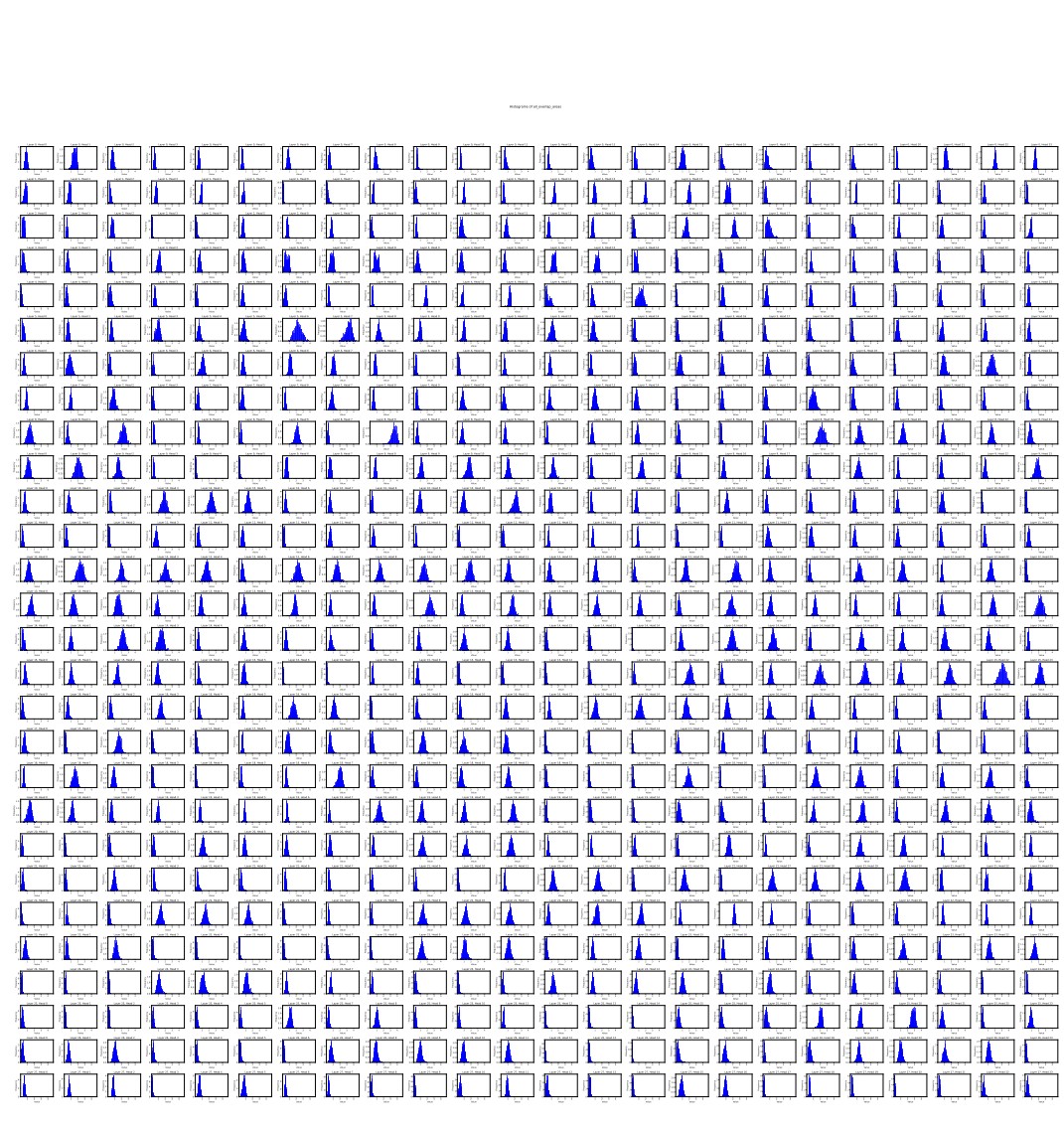

Figure 20: (see on screen) The distribution of RC upper bound (overlap area) for QmSum dataset. The first (last) row corresponds to heads in the first (last) layer.

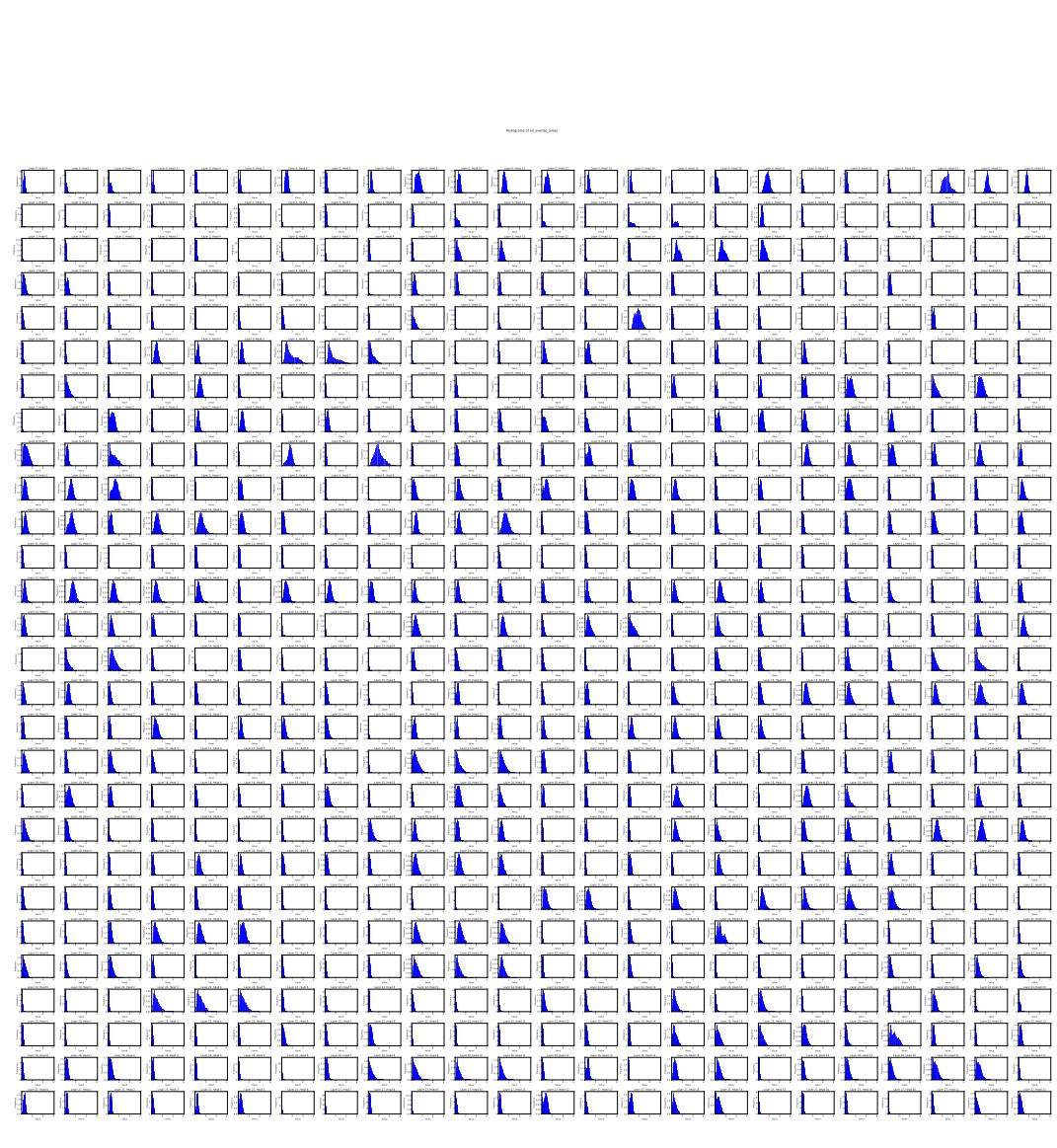

Figure 21: (see on screen) The distribution of RC upper bound (overlap area) for Squad v2 dataset. The first (last) row corresponds to heads in the first (last) layer.

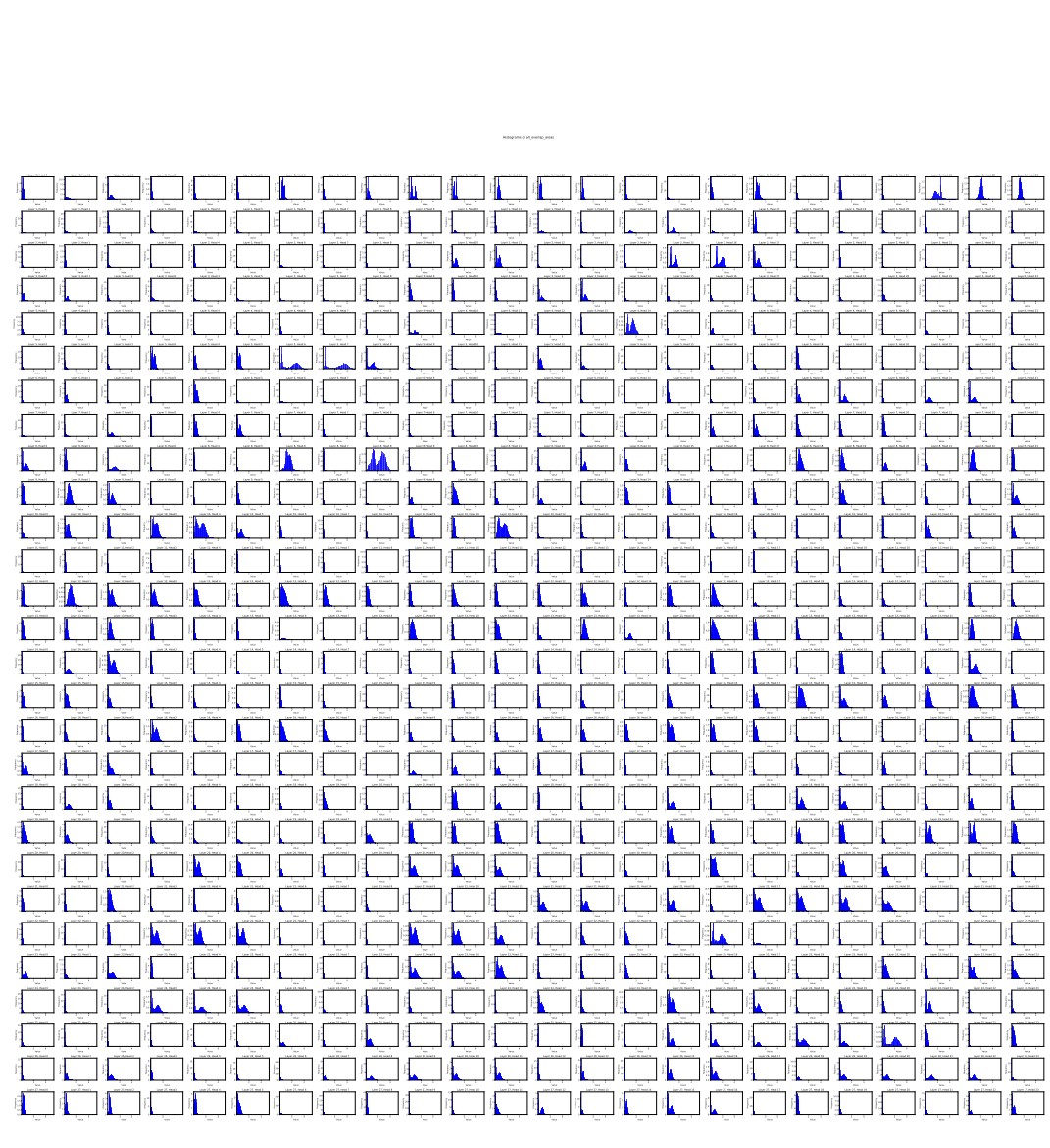

Figure 22: (see on screen) The distribution of RC upper bound (overlap area) for 2WikiMultiHop dataset. The first (last) row corresponds to heads in the first (last) layer.

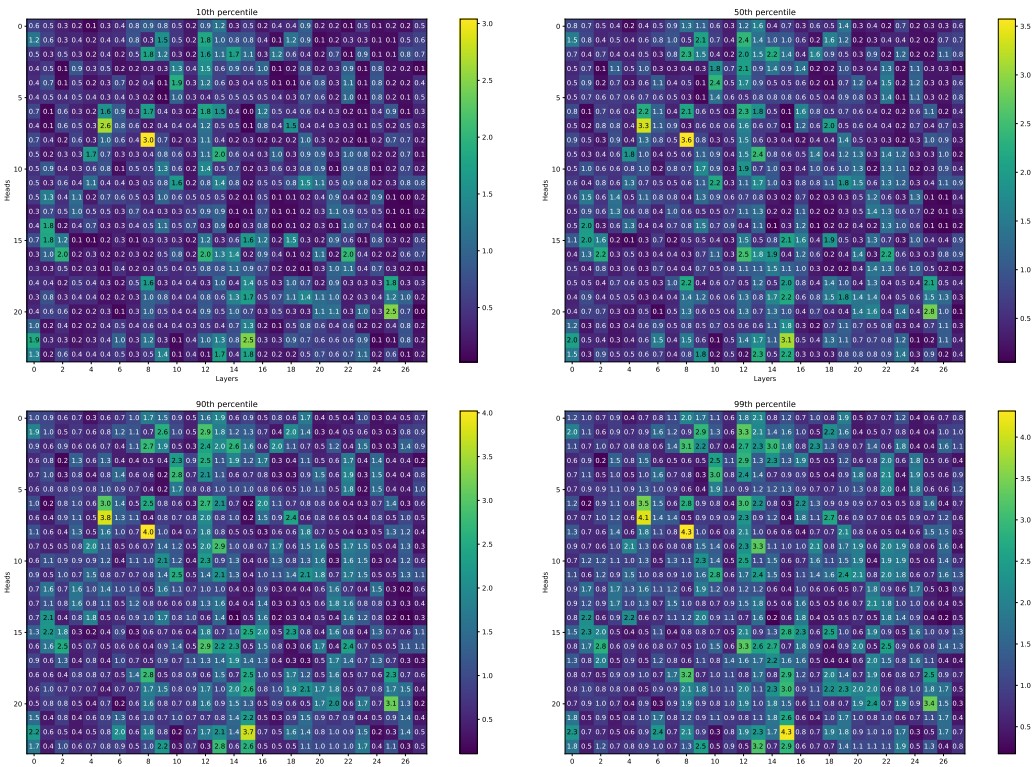

Figure 23: (see on screen) Percentiles of RC upper bound (overlap area) across the QmSum dataset

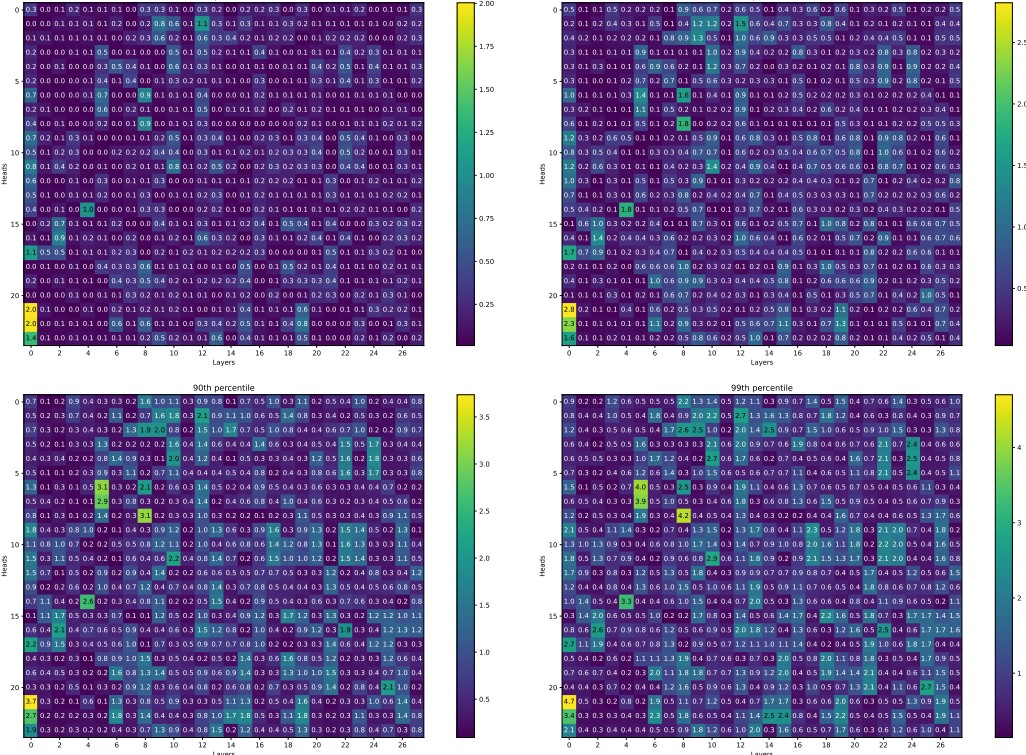

Figure 24: (see on screen) Percentiles of RC upper bound (overlap area) across the Squad dataset

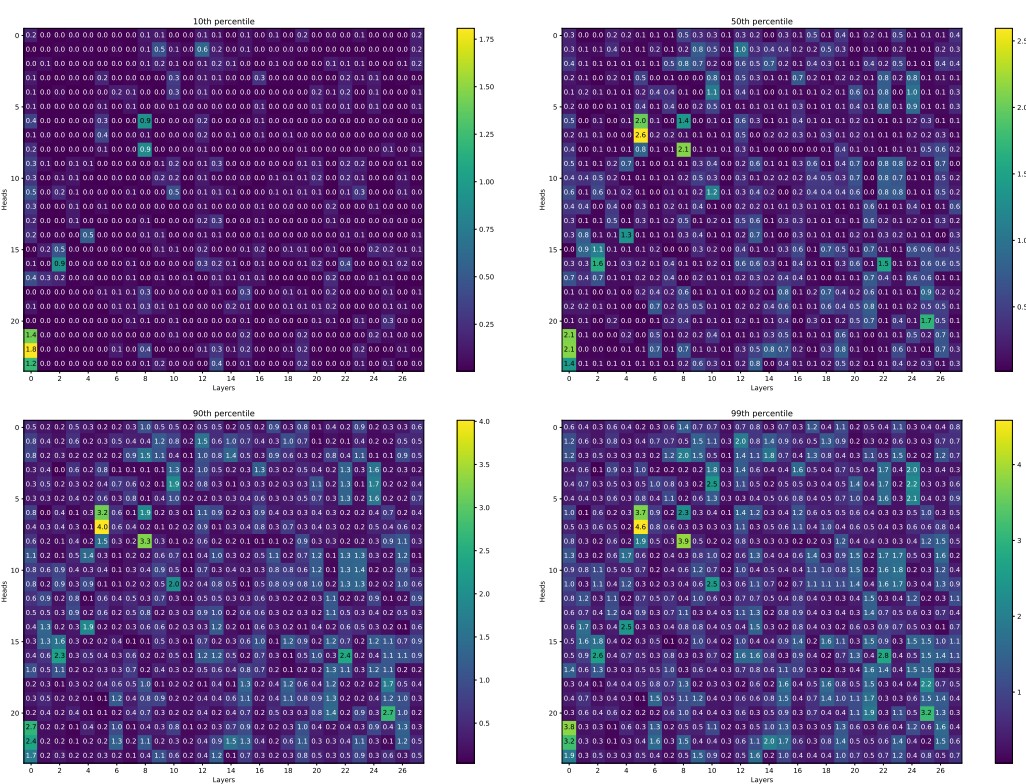

Figure 25: (see on screen) Percentiles of RC upper bound (overlap area) across the 2WikiMultiHop dataset

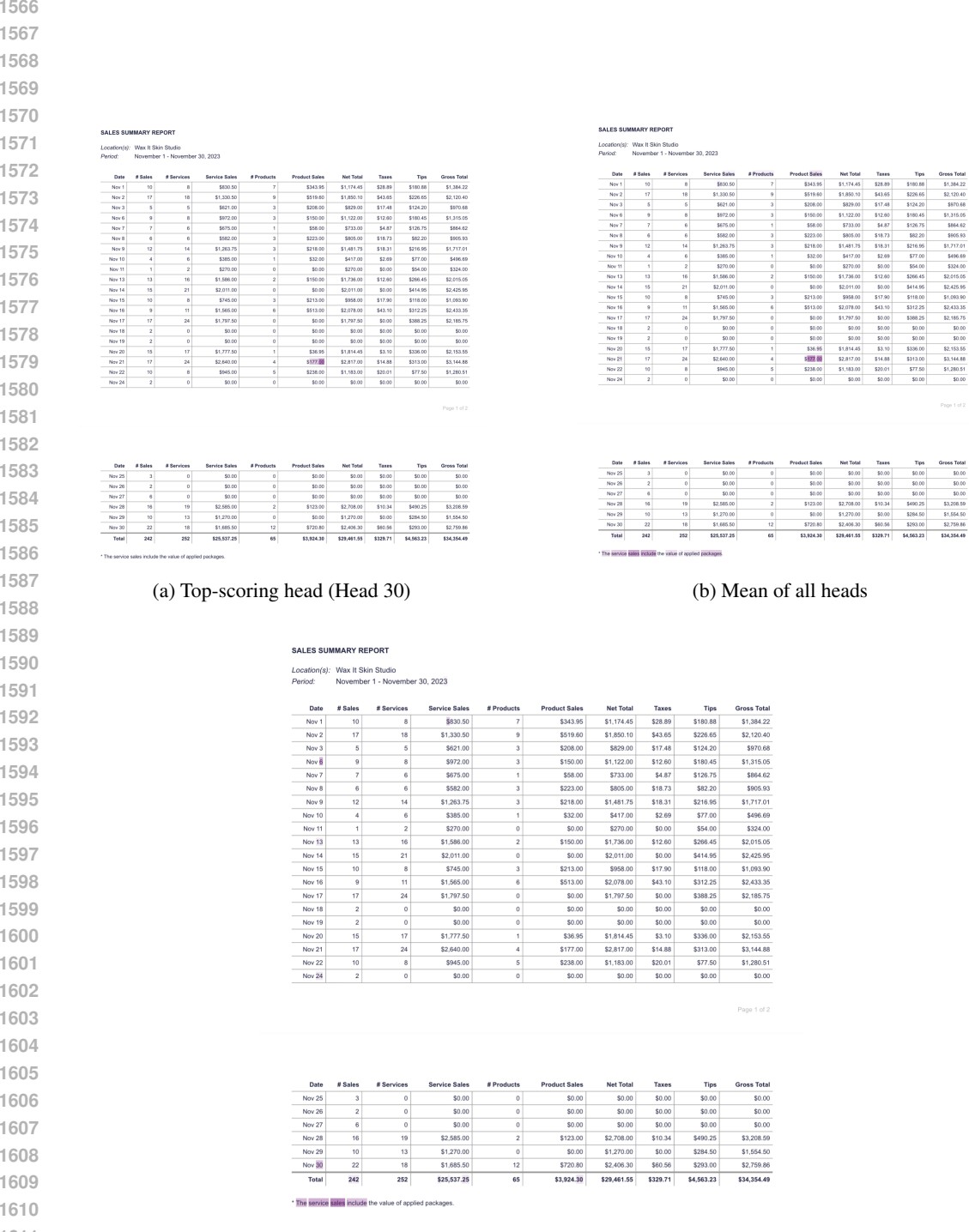

(a) Top-scoring head (Head 30)

(b) Mean of all heads

(c) Worst-scoring head (Head 16)

Figure 26: Attention heatmaps from Layer 15 using three attribution strategies. The top head yields focused and accurate attribution, the mean head shows diluted but somewhat relevant attention, and the worst head highlights largely irrelevant regions.

