# OpenReview forum: "RCStat: A Statistical Framework of Relative Contextualization in Transformers"
_ICLR.cc/2026/Conference — Submitted to ICLR 2026_

### Official Review · Reviewer_mgpv · 2025-10-26

**Soundness:** 2
**Presentation:** 2
**Contribution:** 1
**Rating:** 2
**Confidence:** 4

**Summary:**

The paper introduces RCStat, a statistical framework that leverages pre-softmax attention logits to quantify contextual influence between token groups through a measure called Relative Contextualization (RC). Unlike prior methods relying on post-softmax attention weights—which lose fine-grained relational information due to normalization—RCStat models raw attention logits as random variables and derives an efficient upper bound on their expected difference to estimate contextual relevance. This statistical formalism enables two key applications: adaptive KV-cache compression and token-level attribution, both achieved without retraining or auxiliary supervision.

**Strengths:**

1. The work provides a theoretical justification for their kv cache's importance quantification.

**Weaknesses:**

1. The authors claim "Despite this potential, the usage of pre-softmax attention remains largely underexplored, primarily due to the lack of statistical tools and frameworks to extract structured insights from unnormalized logits." I believe this is not not true. Please check out [1]: a kv cache selection paper that quantifies importance of kv based on pre-softmax scores, or [2].
2. Missing comparison with SOTA baselines such as [1][2].
3. The ultimate goal of most KV-cache compression techniques is inference efficiency (faster decoding, lower memory footprint) with minimal performance loss. While the proposed work emphasizes improved performance under compression (which may itself be questioned in competitiveness to SOTA), the work provides no experiments on efficiency metrics (latency, GPU memory, throughput) - thus its practical viability remains unclear.


[1] Quest: Query-Aware Sparsity for Efficient Long-Context LLM Inference, Tang et. al., ICML 2024.
[2] InfLLM: Training-Free Long-Context Extrapolation for LLMs with an Efficient Context Memory, Xiao et. al., ArXiv.
[3] DuoAttention: Efficient Long-Context LLM Inference with Retrieval and Streaming Heads, Xiao et. al., ICLR 2025.

**Questions:**

1. I'm wondering what is the performance of the proposed method under other long-context tasks such as needle in a haystack?

---

> ### Author Response · Authors · 2025-11-22
>
> Dear reviewer,
>
> We thank you for the thoughtful review. We address each concern.
>
> ---
> # W1: Prior use of pre-softmax attention signals (Quest/InfLLM)
> We acknowledge recent methods, such as Quest and InfLLM, make use of raw attention logits. Our intention was not to claim exclusivity, but to highlight that a general statistical framework, grounded in distributional comparisons, remains missing. We will revise our phrasing to reflect this more precisely. More concretely:
> - Quest/InfLLM use pre-softmax magnitudes heuristically (e.g., top-k, thresholding) to sparsify attention or select tokens.
> - RCStat instead defines random variables (X = Cross-Contextualization, Y = Self-Contextualization) and sets RC = ReLU(X − Y) to quantify how much prompt influence exceeds local influence. RC is thus a distribution-level statistic, not a magnitude filter.
> - We derive a computable upper bound on E[RC] and an efficient estimator (Algorithm 1), enabling token/span/head-level analysis and improving applications like KV eviction and attribution.
>
> We will explicitly clarify that while logits appear in past work, RCStat provides a fundamentally different interpretation and mathematical structure.
> # W2: Comparison with Quest/InfLLM/DuoAttention.
> These methods, while related, are not directly comparable baselines for our use cases:
> - There are practical applications where the CPU memory is a bottleneck such as on-device deployments. In such cases, majority of KV cache has to be evicted during prefill, before generating any auto-regressive decode tokens. Our work contributes to such use cases and the methods that do proper KV eviction, such as TOVA, SnapKV, KNorm etc., are appropriate baselines. In contrast, the goal of Quest and InfLLM is compute reduction, not KV eviction. They retain all KV entries throughout the AR decode phase; although they may offload KV to CPU memory, they do not evict it. We will clarify this distinction in the revision.
> - DuoAttention is an excellent paper that identifies retrieval heads, conceptually similar to selecting high-expected-RC heads in RCStat. However, DuoAttention requires an offline optimization stage on a simulated dataset to determine retrieval vs. streaming heads before deployment. In our production environment, with highly diverse content types and queries, maintaining such simulated datasets and precomputing head roles is infeasible.
>   - RCStat avoids these issues: it requires no offline training or optimization and identifies influential heads on the fly, using context-specific RC scores.
>   - In our KV compression experiments, retrieval vs. streaming behavior is not binary; heads exhibit these properties to varying degrees, and we adjust eviction ratios proportionally.
>   - For attribution, one could use DuoAttention’s retrieval heads, but, as noted in line 431, we focus on inference-only, training-free methods suitable for production deployment.
>
> We will clarify the above distinctions in our revised draft.
> # W3: Inference Efficiency
> This analysis is in Appendix section C.4. We reiterate it here again
> - **Memory savings**: KV memory is the primary bottleneck in real deployments. Compression quality is determined by how much memory can be reduced without harming answer quality. RCStat achieves up to 32% higher memory savings than the best baseline at equal accuracy (Fig. 6c), a substantial improvement.
> - **Computational latency**:
>   - **Prefill (dominant, unchanged)**: Prefill cost is identical across methods and dominates runtime (≈3 ms/layer on Llama-3.2-3B for QMSum with ~1000-token prefixes).
>   - **Compression decision**: RCStat adds 0.68 ms/layer, similar to SNAP (0.66 ms) and TOVA (0.65 ms), and only modestly slower than KNorm (0.13 ms) and STREAM (0.15 ms). Because this step can be pipelined with layer-wise prefill, the effective overhead is negligible.
>   - **Decode**: Decode latency improvements scale with compression ratio. Since RCStat enables more aggressive compression at equal accuracy, it yields proportionally larger decode speed-ups.
> In summary, RCStat provides significantly better memory compression while maintaining latency comparable to SOTA baselines. With an extra page in the camera-ready version, we will move this section into the main paper.
> # Q: Long Context (Needle in a Haystack)
> Although our main experiments do not include long context tasks, RCStat can be applied as follows:
> 1. **KV compression**:
>   - When the question is present during compression, RCStat is naturally effective, since RC highlights context relevant to the query.
>   - When compression must occur before seeing the question, more validation is needed. If memory is not a bottleneck, methods that retain KV (Quest/InfLLM) may be preferable.
> 2. **Attribution**: For post-hoc attribution, RCStat is well-suited. As shown in Fig. 8, only a small set of high-RC heads suffices for maximal attribution accuracy, and RCStat identifies these heads on the fly with no training or optimization.

---

> ### Comment · Reviewer_mgpv · 2025-11-25
>
> Thank the authors for your rebuttal.
>
> 1. The authors state "to highlight that a general statistical framework, grounded in distributional comparisons, remains missing". I think this can never be inferred from "Despite this potential, the usage of pre-softmax attention remains largely underexplored, primarily due to the lack of statistical tools and frameworks to extract structured insights from unnormalized logits" in any ways. The phrase literally says "the usage of pre-softmax attention remains largely underexplored". This shows that you haven't done enough literature survey.
>
> 2. You say "majority of KV cache has to be evicted during prefill, before generating any auto-regressive decode tokens". Doesn't the proposed method also fully compute the kv cache in prefill phase before determining which kv cache to evict? Then how can it relate to your claim of "where the CPU memory is a bottleneck such as on-device deployments"?
>
> 3. I believe the authors can agree with me that the main objective of any KV cache compression or eviction methods is long-context task. Why do we avoid long-context benchmarks here in this case?
>
> Based on these, I would like to maintain my rating.

---

> ### Author Response · Authors · 2025-11-25
>
> We thank the reviewer for responding and highlighting the relevant literature on long-context methods. We would like to address all the three concerns in order:
>
> 1. As noted in our rebuttal, we acknowledge that there are prior works that leverages presoftmax logits, and we therefore stand corrected on our earlier phrasing.
> 2. We would like to clarify that, in production deployment, **we never compute or hold the full KV cache of the entire network at once**. During prefill, before proceeding to layer $l+1$, the KV cache of layer $l$ is evicted. As a result, the peak memory usage never approaches that of storing the full KV cache across all layers.
> 3. As also stated in our rebuttal, in our setting it is infeasible to store KV caches during the decode phase, even in CPU memory, due to device constraints. Consequently, methods such as InfLLM and Quest, while valuable, are not applicable to our use case.
>
> We hope the above clarifications contribute positively to your decision.

---

> > ### Author Response · Authors · 2025-11-25
> >
> > Dear Reviewer,
> >
> > We would also like to emphasize that our paper is not solely about KV cache compression. It also advances the state of the art on the source attribution problem, a task of significant importance for explainability in AI systems. We kindly request that the paper be evaluated in the context of its overall contributions rather than a specific oversight in our related-work discussion. We acknowledge this oversight and will properly cite InfLLM, Quest and DuoAttention in our relevant literature section.
> >
> > We sincerely thank you for bringing it to our attention, which is an essential part of the peer review process.
> >
> > Thank you for your thoughtful consideration.

---

### Official Review · Reviewer_kwdx · 2025-11-01

**Soundness:** 4
**Presentation:** 3
**Contribution:** 4
**Rating:** 8
**Confidence:** 3

**Summary:**

This work proposes a method for quantifying the influence of attention logits by measuring the relative contribution of the prompt part of tokens toward the generation part of the tokens. Basic ideas is to treat the raw attention logits as values for a random variable in a probability density function, and defines two random variables defined for the queries during generation: cross-contextualization which represents the logits for the prompt part of keys and self-contextualization which represents the logits for the generation part of keys. The third variable, relative contextualization, is defined to measure the importance of cross-contextualization over self-contextualization motivated by the focus of prompt during generation. The metric is employed for two tasks: KV cache compression by pruning irrelevant keys and chunk-level prompt attribution, demonstrating superior performance over baselines.

**Strengths:**

- This work presents a novel view of the raw attention logits in order to represent the influence of two parts, i.e., prompt and generation, by introducing random variables, cross-contextualization, self-contextualization and relative-contextualization. Given this view, this work also shows that the upper-bound of expected relative contextualization could be computed by marginization with an efficient algorithm for the computation.
- Experiments are designed systematically with solid results. The KV cache experiments clearly show better tradeoff of the compression and qualities. Chunk-level accuracy for attribution also shows gains when compared with other methods.

**Weaknesses:**

- It takes time to understand the manuscript, given that several key explanation for Equations CC (9) and SC (10) is presented in Appendix. Probably a little bit more intuitive explanation will alleviate the issue.
- Similarly, the chunk-level prompt attribution task is not clearly explained, and thus, readers have to look up the prior studies to understand the setting.

**Questions:**

- $c$ appeared in Equation (SC), but it should be $s$ if my understanding is correct.

---

> ### Author Response · Authors · 2025-11-22
>
> Dear reviewer,
>
> We thank you for seeing the potential of our work and the suggestions for improving the exposition.
>
> ---
> # W: Improving the Exposition
> 1. With an additional page for the main body in the camera-ready version, we are planning to bring the equations 9 and 10 back to the main body and make section 3.1 complete in itself. We will also add more textual explanations for the definitions of Corss Contextulization (CC) and Self Contextualization (SC) to make the reading easier.
> 2. We will describe what is chunk-level prompt attribution in the section 4.2 to make the attribution section self contained.
>
> # Q: Typo
> Thank you for catching the typo in equation SC. It should have been s = p + g, instead it is mistakenly written as s=c+g.

---

### Official Review · Reviewer_kCPH · 2025-11-03

**Soundness:** 2
**Presentation:** 3
**Contribution:** 3
**Rating:** 4
**Confidence:** 3

**Summary:**

The paper introduces RCStat, a method for measuring attribution to prior context by using unnormalized attention weights. It proposes both an exact form and a more efficient approximation of this measure. Then, the paper demonstrates the utility of RCStat by using it to develop both a KV cache eviction method and text attribution method.

**Strengths:**

S1. The idea of RCStat is intriguing-- I like the argument that looking at weights post-softmax loses valuable information. To the best of my knowledge, this is a novel way of measuring influence from prior context.

S2. The formulation of RCStat with separation of context into two contrastable segments is also interesting, and the way that this allows for span-level identification of importance is a nice property.

S3. Attribution is a really hard problem, and one that this paper makes an nice contribution towards. I think the analysis of attribution performance growing worse with adding too many irrelevant attention heads is an interesting one.

**Weaknesses:**

W1. The paper presents a lot of insights, which is nice, but because it takes on three different focuses (deriving RCStat, using RCStat to design a competitive-with-SOTA KV cache eviction strategy, and analyzing RCStat's usefulness for attribution), it sometimes struggles to provide sufficient evidence for each. I organize the rest of the critique into these three subparts.

W2. The initial explanation of RCStat was a bit hard to follow. I found the shapes of density functions drawn in 2(d) to be a bit distracting, and something that it would have been helpful to have clarified earlier-- in particular, why do these density functions overlap only slightly?

W3. To make an argument about applying RCStat as a new KV cache eviction method, you really must address latency in the main text; I think the results in the appendix are somewhat promising, but I don't agree with the statement that this is only "modestly slower" than methods that are 4-5x faster per layer in the compression step (I understand the argument that, compared to prefill+decode, the overall time difference is negligible, but if that is the case you want to make, you should additionally report some metrics of overall time!). ROUGE-1 is not a reasonable quality metric to use; even an older neural metric would make a stronger case. And it seems this is missing several KV cache methods that are prominent in the community -- what about H20 or Pyramid KV? Why is this particular set of baselines the most reasonable choice?

W4. The attribution section is interesting, but I think could benefit from some presentation improvements. Can you report standard deviations and/or significance test the differences between scores in Table 1? L3.1-8B's HS baseline is in a different section from the other L3.1-3B results, but for the Llama 8B and Qwen numbers this is in the same grouping. The pre/post comparison for Llama 8B might be better positioned in a separate table. Are the "least RC" numbers listed for the other models the pre-softmax ones? I was also wishing for a bit more exploration of which heads were good predictors of attribution-- the relevance of depth is mentioned a few times (and discussed in other work), but it would be interesting to hear more about how this works for RCStat's measure of head relevance in particular.

**Questions:**

Q1. Do you have any intuition for why VeriGran performance drops so much more dramatically than QuoteSum performance with increasing number of heads for Qwen in Fig 8b?

Q2. My main critique is that the paper, in trying to do a lot of different things, is not quite doing all of them to a sufficient level. In your conceptualization of this work, do you see the theoretical framing of RCStat itself as the main contribution (with RCStat-based KV cache eviction + RCStat-based attribution being two example applications), or would you place all three on equal footing?

Q3. Like most attention-based methods, this requires explicitly instantiating the attention matrix in memory, instead of calling an efficient kernelized implementation, right? Can you discuss the VRAM requirements imposed by this?

Q4. various specific questions/feedback on the sections, as detailed in the weaknesses above.

Small typos:
- line 463: generalizability misspelled
- line 100: this citation for the phrase mechanistic interpretability seems strange to me -- not clear why this must be cited or that this is the right thing to cite? I'm not as familiar with this literature though so if there is a real reason for this please disregard.
- line 112: "up to 84%" is what that work observed, but this phrasing makes it seem like this is a hard-and-fast maximum, which of course is not true
- line 419: "the an"; more generally I don't really understand what this sentence is trying to say

---

> ### Author Response · Authors · 2025-11-22
>
> Dear reviewer,
>
> We thank you for the detailed review and valuable suggestions.
>
> ---
> # W1 and Q2: What is our main contribution?
> The reviewer’s suggestion to clarify the main takeaway is incisive and strengthens the paper’s overall pitch. We also debated internally whether to demonstrate RCStat on multiple applications or focus on just one.
>
> Since this is the first paper introducing the RCStat framework, we view its general theoretical formulation as the primary contribution. To anchor its real-world utility, we initially explored KV compression. Although our results were strong, the area is crowded, and we felt demonstrating only KV compression might obscure the broader applicability of RCStat. This motivated adding attribution, a less explored but highly valuable task for AI systems, where RCStat’s attention-head prioritization also yields strong performance, even surpassing GPT (Table 1).
>
> Following the reviewer’s suggestion, in the final draft’s introduction, we will emphasize that our main contribution is the theoretical framing of RCstat, and clarify that we show its applicability in two real world problems.
> # W2: Exposition and Figure 2d
> We will add more explanation in Section 3 and clarify that Figure 2d simply illustrates the pre-softmax logit histograms, grouped into cross and self $\langle q,k\rangle$. Figure 1 has a concrete instance of this illustration.
>
> The overlap in Figure 2d actually varies across heads, consistent with the RCscore upper bounds (“overlap area”) in Figures 4b–4d. We will add a figure showing actual histograms for all Llama layers and heads. Figures 23–25 already report overlap-area percentiles across all layer–head pairs.
>
> # W3 and Q3: KV Cache Eviction
> ## Latency
> We thank the reviewer for noting the latency numbers in the Appendix. We will move it from the Appendix to the main paper, now that an extra page will be available.
>
> Higher compression ratios (CR) yield greater memory savings. For a fixed CR, RCStat achieves higher accuracy; for a target accuracy, it supports much higher CR (Table 3; Figures 5–6, 14–19).
>
> Across 10 runs on an A100 (Llama3.1-8B) with a 2048-token input and 128-token decode, average generation time is 4.36 s:
> - 4.10 s decode latency (94%)
> - 0.26 s prefill+compress at 60% compression
>
> RCStat’s compression itself takes only 20 ms (0.004%). All baselines show similar decode latency (~4.1 s), with KNorm having the lowest prefill+compress time (0.245 s).
>
> Prior works (SnapKV, StreamingLLM) show that higher compression reduces context length and thus decode latency. Since RCStat achieves higher ratios at better accuracy (Table 3, ours 70% vs. KNorm’s 40% ), it should in principle yield lower decode latency. However, for short decode lengths (tens of tokens), this reduction is negligible. Thus, RCStat provides far larger memory savings while maintaining comparable latency. We will clarify this in the paper.
> ## Metric
> We agree Rouge is no longer an ideal metric but include it for comparability with prior KV-eviction work.We acknowledged its limitations (Appendix C1, line 402). We also report Value Error Rate (VER), a neural metric measuring final-layer value-vector deviations. VER comparisons (Figures 5a, 5c, 6a, 6c, 14, 15) more accurately reflect degradation relative to no compression.
> ## Baseline
> H2O and PyramidKV (lines 118, 121) are cited, but SnapKV already outperforms them at comparable CR on the same datasets. For brevity, we omitted them as baselines.
> ## VRAM requirements
> RCStat requires only a small query window (8 or 16 tokens) multiplied with the keys (Figures 5 and 6), not the full attention matrix. As noted on line 296, this reduces VRAM to $O(n)$ for n prefill tokens, versus the naive $O(n^2)$ requirement.
> # W4 and Q1: Attribution results
> ## Table 1
> We will reorganize the attribution table by separating HS results from ours for each model. “Least RC” and “Most RC” use pre-softmax scores; for Llama3.1-8B, we additionally show post-softmax RC to illustrate its weaker performance.
> ## Result significance
> Attribution accuracy is computed deterministically per chunk (0/1), with no randomness. We will explicitly report how RCStat matches all correct attributions of the best baseline and captures additional correct cases, demonstrating strict dominance.
> ## Head importance
> Figures 23–25 show percentile heatmaps of overlap areas (upper bounds of expected RC), revealing a consistent pattern: middle-layer heads carry the highest RC scores.
> ## VeriGran vs. Quotesum
> Figure 8b shows, for Qwen3-8B, VeriGran accuracy drops from 92.8% (high-RC heads only) to 39.1% when all heads are included. This is expected:
> - VeriGran is token-level and highly sensitive to noise from low-RC heads.
> - QuoteSum is span-level and thus more robust.
>
> Qwen’s RC signal is concentrated in a small number of heads (long-tail distribution), so adding low-RC heads disproportionately harms fine-grained token attribution.

---

### Official Review · Reviewer_fdAX · 2025-11-03

**Soundness:** 3
**Presentation:** 2
**Contribution:** 2
**Rating:** 4
**Confidence:** 3

**Summary:**

The attention module in Transformer introduces structural bias by sharpening attention toward dominant tokens while flattening the others. It may discard potentially meaningful contextual signals. Corresponding to this phenomenon, the authors posit that logits encode not only what the current layer attends to but also preserve upstream interactions, offering a richer statistical substrate for analysis. Based on this behavior, the authors quantify how different attention heads behave in KV-cache compression and token attribution. By using a newly proposed method, relative contextualization (RC), the authors investigate that most heads have compressible KV-caches, while the few resistant ones provide useful attribution signals. The experimental results on large language models (LLMs) show that the proposed RC-based framework, RCSTAT, can support the improvement of prompt attribution and KV cache reduction in summarization and QA tasks.

**Strengths:**

- The targeting issue of the information loss in attention is fundamental for Transformer-based models, including large language models (LLMs).
- The assumption of the relative contextualization (RC) is based on the actually observed cases.
- The authors provide the computational complexity of the expected RC.
- The experimental results show the effectiveness of RC-based KV cache reduction and prompt attribution on various tasks.
- The authors compared their RC-based method with strong baselines, including state-of-the-art methods.

**Weaknesses:**

- The paper is not self-contained because reading the main text part requires accessing content in appendices like Equations (9) and (10). This is a presentation issue.
- The used models are restricted to small language models that have less than 10B of parameters.
- Runtime comparison is not reported.

**Questions:**

In the KV cache reduction, norms of value vectors are important as well as attention weights shown in the following papers. Could you explain the potential of combining your approach with such methods?
- Alessio Devoto, Yu Zhao, Simone Scardapane, and Pasquale Minervini. 2024. A Simple and Effective L_2 Norm-Based Strategy for KV Cache Compression. In Proceedings of the 2024 Conference on Empirical Methods in Natural Language Processing, pages 18476–18499, Miami, Florida, USA. Association for Computational Linguistics.
- Zhiyu Guo, Hidetaka Kamigaito, and Taro Watanabe. 2024. Attention Score is not All You Need for Token Importance Indicator in KV Cache Reduction: Value Also Matters. In Proceedings of the 2024 Conference on Empirical Methods in Natural Language Processing, pages 21158–21166, Miami, Florida, USA. Association for Computational Linguistics.

---

> ### Author Response · Authors · 2025-11-22
>
> Dear Reviewer,
>
> Thank you for the detailed and constructive feedback! We value the opportunity to address the weaknesses and questions you kindly raised and to further enhance the quality of our work.
>
> ---
>
> # W1: Exposition of the paper
> We thank the reviewer for suggesting the reorganization of section 3.1. With an additional page in the camera ready version, we plan to bring the equations 9 and 10 back to the Main paper, from the Appendix.
>
> # W2: Model sizes
> Our current results validate our method across multiple model families, Qwen and Llama, and model sizes 3B and 8B. Due to limitations of our computational budget, we could not experiment with higher sized models which require many more GPUs. However, our technique has no limitations on its applicability to larger models. In fact, we are planning to open source the code upon acceptance, to enable the research community to analyze the attention heads of bigger sized models using the RCStat framework.
>
> # W3: Run time comparison
> We appreciate the reviewer’s suggestion to analyze inference efficiency. We have provided the analysis in Appendix section C.4 due to space limitations in the main body. We reiterate it here again:
> - **Memory Savings**: Higher KV memory requirement is the primary bottleneck that the compression techniques aim to tackle in real-world deployments. The memory footprint reduction is directly associated with the compression ratio of any KV compression technique. Compression that preserves answer quality, while reducing memory, is a challenging problem. RCStat makes a strong contribution to this front, with significant results, achieving up to 32% higher memory savings than the best performing baseline at the same accuracy levels (derived from Fig. 6c).
> - **Computational latency**: On the computational side, the analysis for any compression technique can be decomposed as:
>   - **Prefill**(unchanged across methods): This step remains unchanged across all methods and dominates runtime (3 ms per layer on Llama-3.2-3B for QMSum with ≈1000 average prefill length).
>   - **Compression decision**: RCStat requires 0.68 ms per layer, which is on par with SNAP (0.66 ms) and TOVA (0.65 ms), while remaining only modestly slower than KNorm (0.13 ms) and STREAM (0.15 ms). Since this step can be pipelined with layer-wise prefill computation, the effective overhead is negligible.
>   - **Decode**: Latency improvements during decoding scale directly with compression ratio. Because RCStat maintains higher answer quality even under more aggressive compression, it enables proportionally greater decoding speed-ups.
>
> To summarise, RCStat not only compresses KV memory more efficiently than prior approaches, but does so with comparable latency to SOTA baselines. With one additional page in the camera-ready version we will move this section to the main body.
>
> # Q1: Importance of Value vectors
> - The first cited paper (Devoto et al.) is one of our baselines in the KV compression experiments, named as KNorm.
> - The norm of the value vector does improve the accuracy of compressed decoding, however, marginally, as can be seen from Table 1 of [Guo et al.’s paper](https://arxiv.org/pdf/2406.12335). The reason lies in the nature of attention operation. In essence, the attention operation is a convex combination of the value vectors, where the combination coefficients are determined by the softmax attention weights. However, the softmax operation is designed to heavily attenuate the majority of the coefficients. Therefore, even without any compression, there are only a few value vectors that contribute towards shaping the value vector of the query token.

---

### Author Response · Authors · 2025-12-02

Dear Area Chair,

We sincerely thank all reviewers for their time and constructive feedback. Across the set of reviews, four important themes emerge: **(1) the novelty and generality of the proposed RCStat framework, (2) strong empirical results and practical utility, (3) Scope and baseline choices, and (4) presentation/exposition issues that are fully addressable** in the camera-ready version. Below we summarize the consensus strengths, clarify points raised in the weaknesses, and explain why the paper is ultimately well-positioned for acceptance.
# 1. Core Contribution is Novel, General, and Well-Supported
Reviewers fdAX, kCPH, and kwdx highlight the novelty and importance of our central idea: modeling raw attention logits statistically through Cross-, Self-, and Relative Contextualization (RC). This formulation provides:
- a **distributional framework** for interpreting contextual influence,
- an **efficient upper bound** enabling practical RC computation, and
- more informative signals at multiple granularities, **head-, token-, and span-level**, than that of post-softmax attention weights.

Even reviewer mgpv acknowledges the value of our theoretical justification for KV importance. We clarified that while prior work (Quest, InfLLM) also uses logits, they operate via heuristic magnitude filtering, whereas RCStat provides a principled statistical formulation, not a heuristic. Unfortunately in spite of acknowledging these strengths and our detailed clarifications, the negative verdict was not updated. We are unclear if any further clarifications were expected.
# 2. Strong Experimental Evidence Across Two Distinct Tasks
Reviewers kwdx and kCPH recognize the experiments as systematic, solid, and insight-yielding. A **key revelation**, enabled by Expected RC measurements, is that **the importance of attention heads remain similar across tasks and follow a power law**. This insight drives our adaptive head-wise KV compression and explains why using fewer, high-RC heads improves attribution accuracy.
## KV Cache Compression
RCStat consistently delivers higher memory savings than strong baselines (SnapKV, KNorm, TOVA), achieving up to 32% more reduction at equal accuracy. Reviewers fdAX, kCPH, and kwdx find these conclusions sound. We also addressed concerns about efficiency:
- Latency: RCStat adds only 20 ms per 2048-token prefill on Llama-8B (<0.5% of total generation time).
- Decode speed: Since all methods share identical decode latency, RCStat’s higher compression enables faster decoding for long sequences.

Per reviewer kCPH’s suggestion, and with the extra page, we will move the full latency analysis (not just microbenchmarks) into the main paper.
## Attribution
Reviewer kCPH agrees that attribution is difficult, underexplored, and important. RCStat achieves strong performance, often surpassing GPT-based heuristics, and uncovers a stable pattern of head-level importance. Reviewer kCPH finds these results “interesting” and “intriguing,” requesting only clearer presentation, which we will address through improved tables, explanations, and significance discussion.

# 3. Baseline Choices, Scope and Model Sizes Are Reasonable
We address kCPH’s question about H2O and PyramidKV by noting that SnapKV, already included as a baseline, outperforms both on the same datasets, making their omission appropriate. Reviewer mgpv’s concerns centered on framing and positioning, which we resolved in the rebuttal. We clarified that our scope is KV cache eviction, where Quest and InfLLM are not appropriate baselines because they target multi-level KV cache management, not eviction.

We validated RCStat across two model families (Llama, Qwen) and both 3B and 8B variants, already substantial given the computational cost of GPU usage. Multiple reviewers accepted this limitation as reasonable. The method itself is model-size agnostic, and we commit to open-sourcing all code, for the community to validate on other model families and sizes.

# 4. Presentation Concerns Are Fixable and Already Addressed
Some reviewers noted that some exposition required consulting the appendix. With the extra page, we will:
- move Equations (9) and (10) into the main text,
- add intuitive explanations for CC/SC/RC and Figure 2, and
- make the attribution-task description fully self-contained.
These issues are purely cosmetic and do not affect the correctness or value of the contributions.
# Concluding Reflections
No reviewer challenges the correctness or fundamental validity of RCStat; all critiques pertain to presentation or scope, which are straightforward to fix. Multiple reviewers explicitly note that the approach is principled, general, and novel, with meaningful contributions to both KV compression and attribution.

Given the strong positive sentiment, the absence of substantive methodological concerns, and the fact that all requested clarifications have been addressed, we respectfully submit that RCStat is well-positioned for acceptance.

---

### Meta-Review · Area_Chair_BaZR · 2026-01-06

**Summary:**

The paper proposes RCStat, a statistical framework over pre-softmax attention logits that defines cross-, self-, and relative contextualization and derives an efficient upper bound for expected RC. It applies this signal to KV-cache eviction and prompt attribution, with experiments on Llama and Qwen (3B/8B). Reviewers appreciate the principled use of logits and the dual applications. Main concerns are incomplete positioning versus prior logit-based methods (Quest, InfLLM, DuoAttention), limited scale (≤8B) and lack of long-context benchmarks, and insufficient end-to-end efficiency reporting in the main text. Presentation issues are viewed as fixable.

**Reviewer Concerns:**

Addressed by rebuttal:

Clarified novelty claim: RCStat is a distributional framework rather than a magnitude heuristic, with an efficient estimator and head/span analysis.

Efficiency details: Provided latency breakdowns, decode-time arguments, and VRAM discussion; committed to moving these to the main text.

Baselines rationale: Argued eviction vs compute sparsity distinction; explained omission of H2O/PyramidKV given SnapKV’s strength.

Exposition fixes: Commit to moving key equations to the main paper and clarifying the attribution task; corrected a typo in SC.

Outstanding:

Literature framing remains too strong in the current draft; must explicitly acknowledge prior pre-softmax work and clearly delineate scope.

Empirical scope: No standard long-context benchmarks; models capped at 8B. This limits external validity for KV compression.

End-to-end efficiency: Claims are plausible, but wall-clock throughput and decode-speed comparisons at matched quality should be in the main text, not only the appendix.

Attribution reporting: Needs clearer table organization and explicit significance reporting.

**Reviewer Scores:**

Reviewer advocating accept (high score): Likely unchanged.

Borderline reviewer focused on clarity/latency/attribution presentation: May stay borderline or increase slightly given planned fixes.

Borderline reviewer focused on scope/baselines: Likely unchanged without added long-context and expanded baselines in tables.

Critical reviewer citing prior work and long-context gaps: Likely unchanged.

---

### Decision · Program_Chairs · 2026-01-26

Reject